# Hepatic neddylation deficiency triggers fatal liver injury via inducing NF-κB-inducing kinase in mice

Cheng Xu[1], Hongyi Zhou[1], Yulan Jin[2], Khushboo Sahay[1], Anna Robicsek[1], Yisong Liu[1], Kunzhe Dong [3], Jiliang Zhou [3], Amanda Barrett[2], Huabo Su [4] & Weiqin Chen [1] ✉

The conjugation of neural precursor cell expressed, developmentally down-regulated 8 (NEDD8) to target proteins, termed neddylation, participates in many cellular processes and is aberrant in various pathological diseases. Its relevance to liver function and failure remains poorly understood. Herein, we show dysregulated expression of NAE1, a regulatory subunit of the only NEDD8 E1 enzyme, in human acute liver failure. Embryonic- and adult-onset deletion of NAE1 in hepatocytes causes hepatocyte death, inflammation, and fibrosis, culminating in fatal liver injury in mice. Hepatic neddylation deficiency triggers oxidative stress, mitochondrial dysfunction, and hepatocyte reprogramming, potentiating liver injury. Importantly, NF-κB-inducing kinase (NIK), a serine/ Thr kinase, is a neddylation substrate. Neddylation of NIK promotes its ubiquitination and degradation. Inhibition of neddylation conversely causes aberrant NIK activation, accentuating hepatocyte damage and inflammation. Administration of N-acetylcysteine, a glutathione surrogate and antioxidant, mitigates liver failure caused by hepatic NAE1 deletion in adult male mice. Therefore, hepatic neddylation is important in maintaining postnatal and adult liver homeostasis, and the identified neddylation targets/pathways provide insights into therapeutically intervening acute liver failure.

Liver diseases pose a significant worldwide health problem. Rapidly progressive liver failure (a.k.a. acute/subacute liver failure, ALF) can occur after acute liver injury induced by drugs, toxins, viral infections, etc. ALF is a dramatic clinical syndrome in which a previously normal liver fails within days or weeks due to a massive loss of parenchymal cells[1,2]. ALF accounts for 2% of liver mortality. Therapy for ALF has been limited to date, with liver transplantation as the only proven life-saving intervention[1,2]. Other interventions that aim to reconstitute liver functions have generally been ineffective in reducing ALF mortality. Various mechanisms responsible for hepatotoxicity have been

proposed, including oxidative stress, disruption of mitochondrial function, alteration of $Ca^{(2+)}$ homeostasis, etc[3]. Meanwhile, an acute injury may render the liver unable to regenerate or to do so sub-optimally, attributing to the pathological conversions of hepatocytes to biliary epithelial cell (BEC)/progenitor lineage cells[4]. The mechanisms underlying acute liver failure are not fully understood, especially at the post-translational level.

NEDD8 (neural precursor cell expressed, developmentally down-regulated 8) is an 81-amino-acid ubiquitin-like protein. Neddylation attaches one or a chain of NEDD8 moieties to lysine residues of target

[1]Department of Physiology, Medical College of Georgia at Augusta University, Augusta, GA 30912, USA. [2]Department of Pathology, Medical College of Georgia at Augusta University, Augusta, GA 30912, USA. [3]Department of Pharmacology and Toxicology, Medical College of Georgia at Augusta University, Augusta, GA 30912, USA. [4]Vascular Biology Center, Medical College of Georgia at Augusta University, Augusta, GA 30912, USA. ✉e-mail: wechen@augusta.edu

proteins. This process is mediated by the sequential action of the NEDD8 activating enzyme E1 (NAE), NEDD8-conjugating enzyme E2 (UBC12 and UBE2F), and certain NEDD8-E3 ligase(s)[5]. NAE is a heterodimer comprised of the regulatory subunit NAE1 and the catalytic subunit ubiquitin-like modifier activating enzymes 3 (UBA3)[6]. A compound, Pevonedistat (MLN4924, MLN) can abolish NAE-mediated NEDD8 activation and thus inhibit neddylation[7]. Dysregulation of neddylation has been implicated in various pathological conditions including liver fibrosis[8] and hepatocellular carcinoma (HCC)[9, 10]. Inhibition of neddylation by MLN was shown to ameliorate liver fibrosis[8], steatosis[11], and cancer[9, 10] in preclinical mouse models. Despite its relative tolerability in treating multiple malignancies[12–14], preclinical investigations found that MLN upregulated hepatic transaminases[13] and reduced the activation threshold for TNF-mediated hepatic cell death[15]. Recently, mice with embryonic-onset hepatocyte-specific deletion of NEDD8 or the catalytic subunit UBA3 were shown to exhibit neonatal death, which was primarily attributed to hepatic cellular senescence[16]. The role of the neddylation pathway in liver pathophysiology remains incompletely explored.

Cullins are the best-described substrates of neddylation, and neddylation of cullins activates Cullin-RING E3 ubiquitin ligases (CRLs), which are responsible for ≈20% of cellular protein degradation via the proteasome[17]. Recent research uncovered various non-cullin targets of neddylation that participate in an array of biological processes including transcription, autophagy, and signaling[5, 18]. NF-κB-inducing kinase (NIK), also called MAP3K14, is a constitutively active Ser/Thr kinase, and its levels are very low in quiescent cells due to rapid ubiquitination and proteasome-mediated degradation[19, 20]. Stabilized NIK phosphorylates and activates IKKα, which in turn phosphorylates p100, triggering its proteolytic cleavage to generate the mature p52 form of the non-canonical NF-κB2[20]. Hepatic NIK is aberrantly activated in both mice and humans with chronic liver diseases, and mouse hepatocyte overexpression of NIK triggers lethal liver failure[21, 22]. However, whether neddylation regulates NIK stability and activation, and thus liver function, is completely unknown.

The present study revealed that hepatic-specific blockade of neddylation by NAE1 deletion activates oxidative stress, impairs mitochondrial metabolism, and induces excessive hepatocyte reprogramming. Importantly, we identified NIK as a NEDD8 target. Neddylation inhibition induces aberrant NIK expression and activation, which contributes to the development of ALF. Furthermore, N-acetylcysteine (NAC) therapy at least partially rescued NAE1 deficiency-induced ALF. Thus, neddylation is required for maintaining postnatal and mature hepatocyte function and fate.

## Results

### Embryonic-onset deletion of NAE1 in hepatocytes causes liver dysfunction and premature death in mice

The relevance of neddylation to human ALF is poorly defined. Analysis of two existing Microarray datasets [GSE38941[23] and GSE98651[24]] of human HBV-associated ALF patients revealed significant perturbations of 13 transcripts encoding neddylation enzymes. Especially, the mRNA expression of *NAE1* was significantly downregulated in the ALF liver of both datasets (Supplemental Fig. S1a, b). This suggests that the neddylation pathway is perturbed in HBV-associated ALF.

To test *Nae1* gene function in vivo, we generated *Nae1^LKO* (LKO) mice with hepatic deficiency of neddylation by breeding *Nae1^f/f* with *Alb-Cre^+* mice. *Nae1^f/f* mice were used as wild-type (WT) controls (Fig. 1a). As expected, NAE1 expression was markedly reduced from postnatal (P) 7 until P14 in the livers of LKO mice. Deletion of NAE1 caused marked downregulations of global neddylation-conjugated (N8-Conj.) proteins and neddylated cullins (N8-CULs), such as CUL2 especially in P7 and P10 LKO livers (Fig. 1a). LKO mice were born at the expected Mendelian ratio and were outwardly normal with

comparable body weights until P14. However, starting from P14, they displayed growth retardation (Fig. 1b) and increased mortality rates, with only 40% of mice surviving by 8 weeks of age (Fig. 1c). Hepatomegaly (Fig. 1d), mild hypertriglyceridemia (Supplemental Fig. S1c), and severe hypercholesterolemia (Supplemental Fig. S1d) were evident as early as P10 in male LKO mice. At P14, LKO mice displayed hypoglycemia (Supplemental Fig. S1e), and exhibited progressive development of fluid-filled cysts, with multiple cysts throughout the liver by P21 (Fig. 1e). This was in agreement with the dramatically upregulated plasma levels of alanine transaminase (ALT) and bilirubin (Fig. 1f), indicative of hepatocyte damage and impaired liver function.

Histologically, the male P14 LKO liver displayed azonal, patchy apoptosis shown as acidophil bodies, and very reactive-looking ductal structures with lots of nuclear atypia and cytological irregularity, suggestive of duct injury associated with ductular proliferation (Fig. 1g). There were occasional foci of confluent and coagulative necrosis and massive immune cell infiltration, indicating hepatocyte cell death and inflammation. Pathological scoring confirmed moderate portal and lobular inflammation in P14 LKO livers with no obvious steatosis (Fig. 1g). Sirius red staining identified the presence of periportal fibrosis with early bridging and pericellular fibrosis (stage 1) in the livers of male P14 LKO mice (Fig. 1g). qRT-PCR analysis revealed increased expression of leukocyte marker genes (*Lgals3*, *Cd68*), and pro-inflammatory cytokines and chemokines (*Ccl2* and *Cxcl14*) (Fig. 1h). Fibrosis was further validated by increased or a trend toward increased expression of fibrotic genes (*Ccn2, Acta2, Col1a1, Col3a1*) (Fig. 1i) as well as elevated liver hydroxyproline contents (Fig. 1j). Female LKO mice displayed even earlier-onset hypertriglyceridemia, hypercholesterolemia, and more severe fibrosis score along with a higher mortality rate (Supplemental Fig. S1f–i). Together, these data suggest that an embryonic-onset hepatic neddylation deficiency leads to ALF with unique cystogenesis, partially recapitulating the features of liver injury/inflammation/fibrosis cascade observed in human ALF.

### Adult-onset, hepatocyte-specific deletion of NAE1 causes lethal liver injury

To study the role of hepatocyte neddylation in adult mice, we injected adult *Nae1^f/f* mice with AAV8-TBG-Cre virus to induce hepatic NAE1 deletion (hereafter, AAV-Cre); AAV8-TBG-eGFP virus was used as a control (hereafter, AAV-GFP) (Fig. 2a). An efficient deletion of NAE1 was verified by the reduced expression of NAE1 and neddylated conjugates (such as CUL1) in livers as early as 10 days (D10) post-AAV-Cre injection (Fig. 2a). Body weights of AAV-Cre mice were ~40% lower at D24, whereas livers of AAV-Cre mice were significantly enlarged at D21 (Fig. 2b). AAV-Cre mice developed jaundice but no cysts by D24 (Fig. 2c and Supplemental Fig. S2a). Not surprisingly, starting from D21 post-injection, AAV-Cre mice exhibited higher levels of plasma ALT (Fig. 2c), whereas markedly elevated levels of plasma bilirubin (Fig. 2c), triglyceride (TG) (Supplemental Fig. S2b), cholesterol (Supplemental Fig. S2c), and hypoglycemia (Supplemental Fig. S2d) were observed until D24 post-injection in these mice compared with AAV-GFP mice. All AAV-Cre mice died within 30 days after injection (Fig. 2d). In D24 AAV-Cre mice, liver architectures were completely disrupted, with azonal apoptosis and a centrilobular (acinar zone 3) necrosis together with marked infiltration of immune cells, indicative of damaged hepatocytes and inflammation (Fig. 2e). Pathological scoring revealed moderate portal and lobular inflammation in D24 AAV-Cre livers (Fig. 2e). There was also steatosis in zone 3 of the liver, predominantly microsteatosis with occasional macrosteatosis, which was confirmed by intensified Oil Red O staining and elevated liver TG contents in D24 AAV-Cre livers (Supplemental Fig. S2e, f). Sirius red staining revealed significant periportal fibrosis with bridging and pericellular fibrosis (stage 2) in D24 AAV-Cre livers compared to that

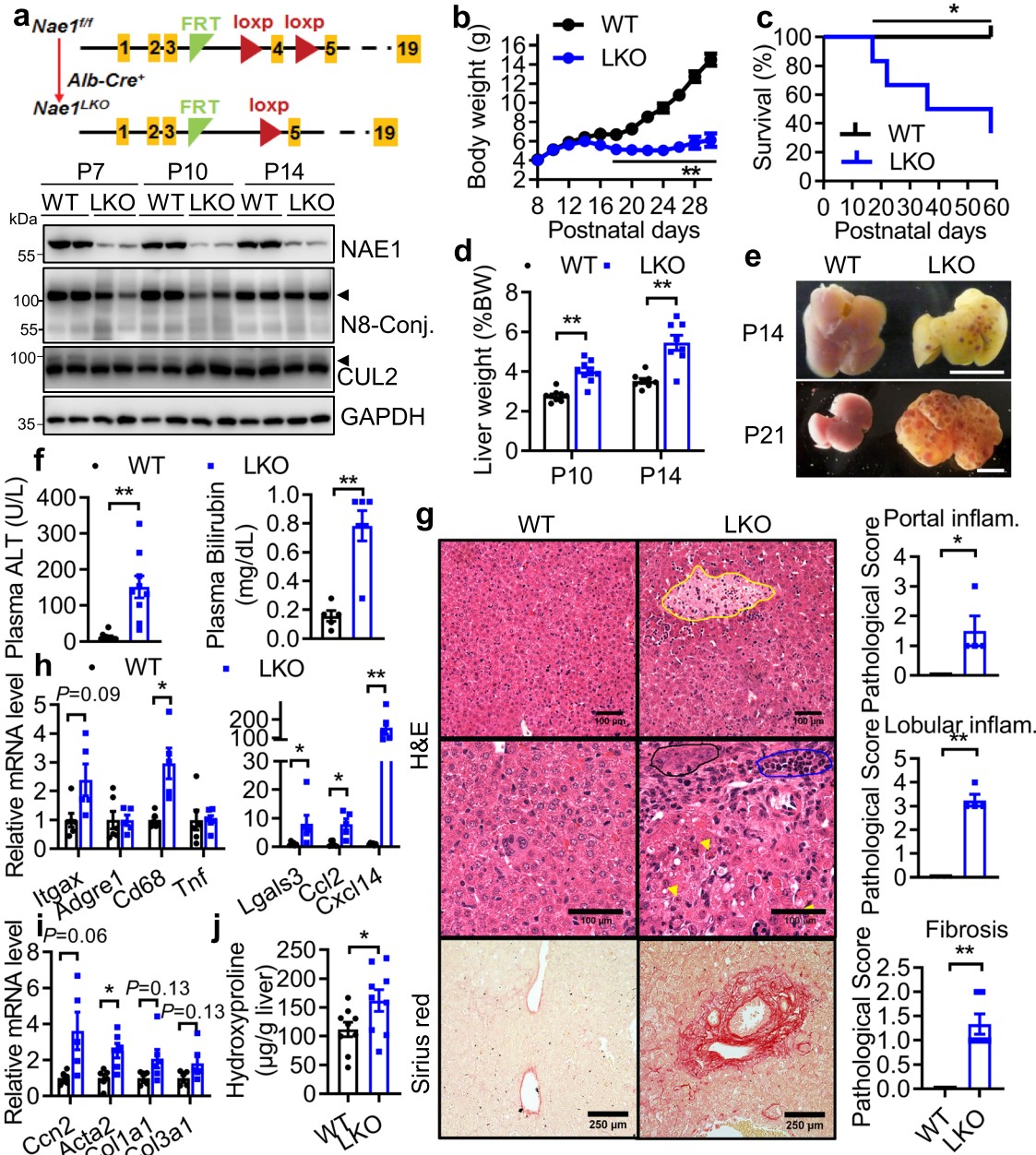

**Fig. 1 | Embryonic-onset, hepatocyte-specific deletion of NAE1 causes liver failure and premature death. a** Schematic diagram depicting the generation of *Nae1$^{f/f}$; Alb-Cre +* (LKO) mice with embryonic-onset hepatocyte-specific deletion of NAE1 by crossing *Alb-Cre +* mice with *Nae1* floxed (*Nae1$^{f/f}$*) mice. *Nae1$^{f/f}$* mice served as wild-type (WT) controls. Western blot of NAE1 and neddylated conjugates (N8-Conj.) in liver extracts of WT and LKO mice at P7, P10, and P14 Arrowhead indicates neddylated CULs and CUL2. Two independent experiments. **b** BW of WT and LKO mice till postnatal 30 days (*n* = 6 per group). **P < 0.005. **c** Survival curves of WT and LKO mice (*n* = 6 per group). Log-rank (Mantel-Cox) test. *P = 0.0183. **d** Liver weight as normalized to BW at P10 (WT: *n* = 9; LKO: *n* = 9, **P = 0.000043) and P14 (WT: *n* = 8; LKO: *n* = 8, **P = 0.000275). **e** Gross morphology of livers at P14 and P21. Scale bar: 25 mm. **f** Plasma ALT (WT: *n* = 10; LKO: *n* = 9, **P = 0.0002) and bilirubin (WT: *n* = 5; LKO: *n* = 6, **P = 0.0006) levels of WT and LKO mice at P14. **g** H&E and

Sirius red staining of liver sections, pathological scores for portal (*n* = 4 per group, *P = 0.024) and lobular (*n* = 4 per group, **P < 0.0001) inflammation as well as fibrosis based on Sirius red staining (*n* = 6 per group, **P < 0.0001) at P14. Yellow circle: the necrotic area; Yellow arrowhead: acidophilic body; Black circle: ductular proliferation; Blue circle: infiltrating neutrophils. Scale bar: 100–250 μm.
**h–i** Relative mRNA levels of inflammatory and fibrotic genes in livers of P14 WT and LKO mice as determined by qRT-PCR (*n* = 6 per group). *P < 0.05, **P < 0.005. **j** Liver hydroxyproline contents as normalized to tissue weights in P14 WT and LKO mice (*n* = 9 per group). *P = 0.024. Male mice were used. Multiple unpaired *t*-tests with the Holm-Sidak method in (**b, d, h, i**). Unpaired *t*-test, two-tailed in (**f, g,** and **j**). WT (black), LKO (blue). All quantitative data were presented as mean ± SEM. Source data are provided as a Source Data file.

of AAV-GFP mice (Fig. 2e). The increased inflammation and fibrosis in AAV-Cre livers was further supported by the upregulated expression of genes related to inflammation and fibrosis (Fig. 2f). Hydroxyproline contents were also elevated in D24 AAV-Cre livers (Fig. 2g). Together, these data suggest that hepatocyte-specific deletion of NAE1 in adult mice causes fatal liver failure.

**Hepatic neddylation deficiency promotes hepatocyte damage and excessive fetal liver reprogramming**

We next examined hepatocyte damage in the livers of D24 AAV-Cre mice. Evans blue dye (EBD) staining, an indicator of necrosis/necroptosis[25], was remarkably increased in NAE1-deficient livers (Fig. 3a). TUNEL staining revealed more internucleosomal DAPI-

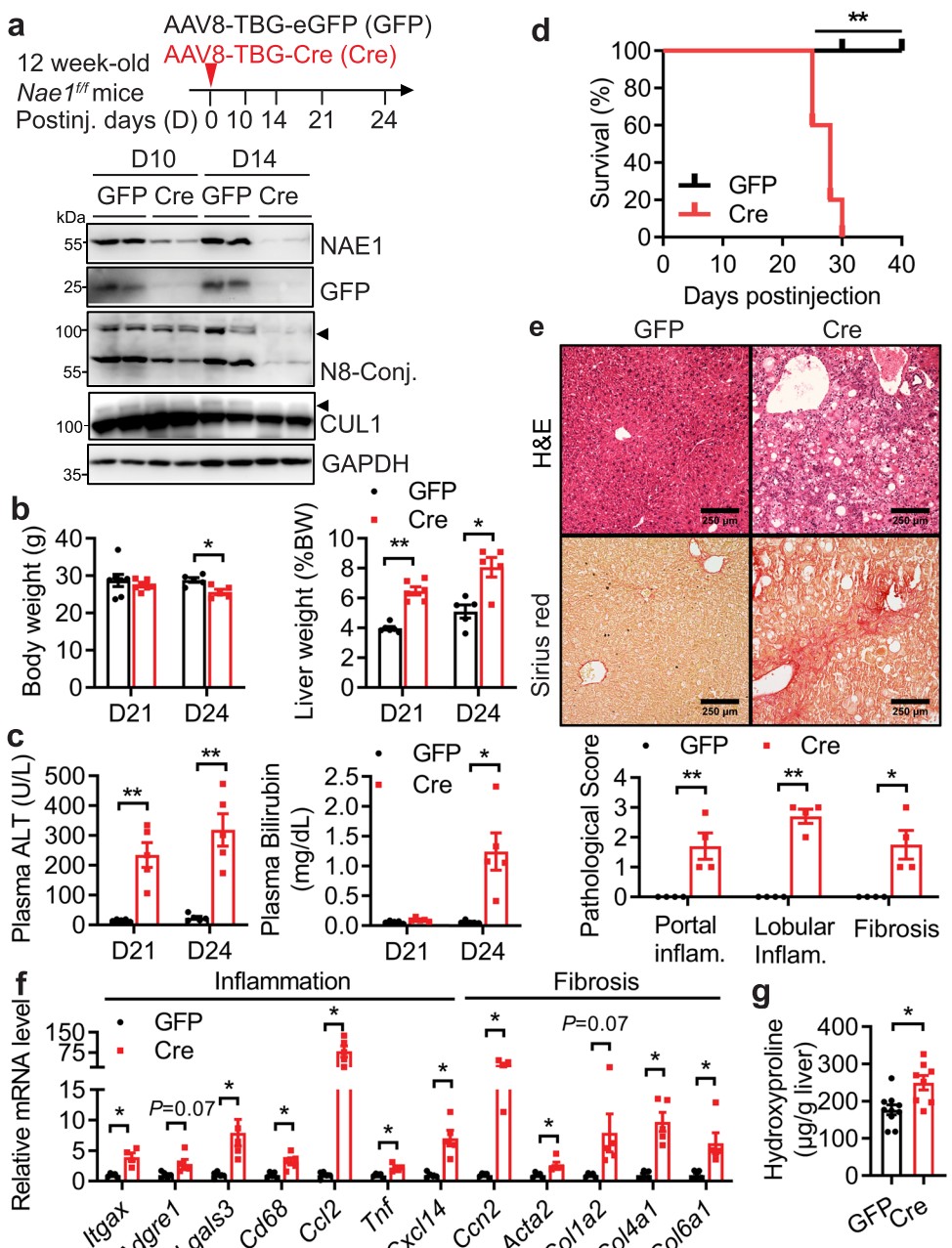

**Fig. 2 | Adult-onset, hepatocyte-specific deletion of NAE1 causes liver failure.**
**a** Schematic diagram depicting intravenous injection of AAV8-TBG-Cre (AAV-Cre, Cre) to 12-week-old *Nae1^f/f* mice to induce NAE1 deletion in mature hepatocytes. AAV8-TBG-eGFP served as a control (AAV-GFP, GFP). NAE1, GFP, and neddylated conjugates (N8-Conj.) were analyzed by Western blot in liver extracts of GFP and Cre mice at D10 and D14 post-virus injections. Arrowheads indicate neddylated CULs and CUL1. Representatives of three independent experiments were shown. **b** BW and liver weight normalized to BW of GFP and Cre mice at D21 (*n* = 6 per group) and D24 (*n* = 5 per group). **c** Plasma ALT and bilirubin levels of GFP and Cre mice at D21 and D24 post-virus injections (*n* = 5). Unpaired *t*-tests, two-tailed in **b**, **c**. *P* < 0.05, **P* < 0.005. **d** Survival curves (GFP: *n* = 6; Cre: *n* = 5). Log-rank (Mantel–

Cox) test. **P* = 0.0008. **e** Representative images of H&E and Sirius red staining of liver sections, with pathological scores assessed for the portal (**P* = 0.016), lobular (***P* = 0.000085) inflammation, and fibrosis (**P* = 0.016) (*n* = 4 per group). Scale bar: 250 μm. **f** Relative mRNA levels of inflammatory and fibrotic genes in livers as determined by qRT-PCR (*n* = 5 per group). **P* < 0.05. **g** Liver hydroxyproline contents as normalized to tissue weights (GFP: *n* = 10; Cre: *n* = 8). **P* = 0.0053. All experiments in **e–g** were performed in the livers of male GFP and Cre mice at D24 post-virus injections. Multiple unpaired *t*-tests with the Holm-Sidak method in (**e**, **f**). Unpaired *t*-tests, two-tailed in (**g**). GFP (black), Cre (red). All quantitative data were presented as mean ± SEM. Source data are provided as a Source Data file.

stained and cytoplasmically localized [derived from postnecrotic DNA hydrolysis[25]] TUNEL-positive cells in AAV-Cre livers (Fig. 3b). Consistently, cleaved caspase-3 and caspase-8, together with the cardinal necroptotic regulators, such as MLKL, phospho-RIPK3, and total RIPK3, were drastically upregulated in livers of D21 and D24 NAE1-deleted AAV-Cre mice (Fig. 3c). Immunohistochemistry of RIPK3 further confirmed the necroptosis was more concentrated in zone 3 of

AAV-Cre livers (Supplemental Fig. S3a). Furthermore, D24 AAV-Cre livers were almost devoid of mRNA transcripts of mature hepatocyte marker genes (e.g. *C/ebpα*, *Alb*, *Hnf4α*, and *Esrp2*) (Fig. 3d). The downregulation of C/EBPα in AAV-Cre livers was confirmed at the protein level (Fig. 3c). In contrast, the mRNA transcripts of BEC/progenitor marker genes[26], such as *Cd34*, *Sox9*, *Krt19*, *Opn*, and *Pkm*, were significantly increased in D24 AAV-Cre livers (Fig. 3d), suggesting

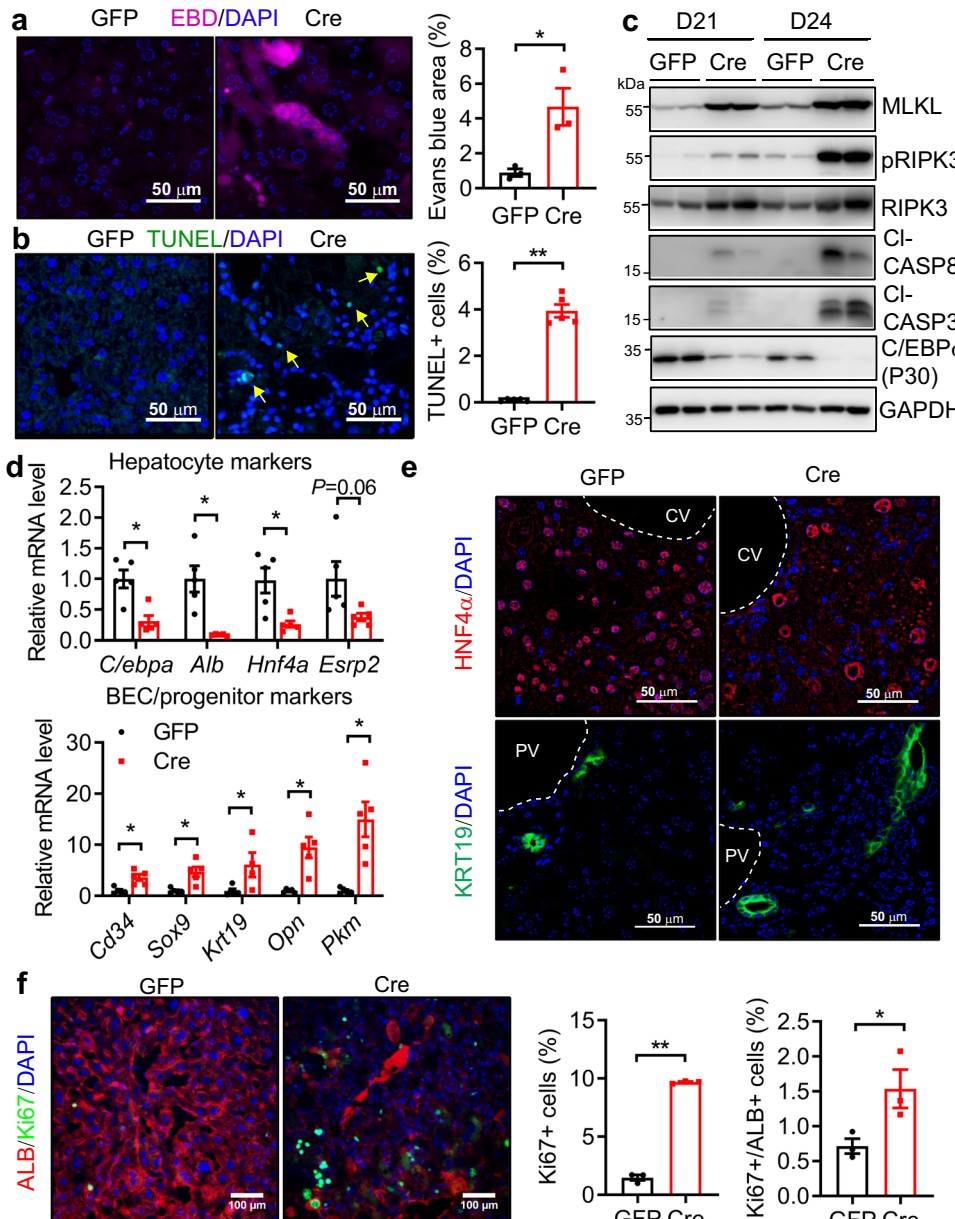

**Fig. 3 | Hepatic neddylation deficiency promotes hepatocyte damage and excessive fetal liver reprogramming. a** Representative EBD staining and quantification of total EBD⁺ areas (*n* = 3 per group, **P* = 0.0255); **b** representative TUNEL labeling and quantification of TUNEL⁺ cells (*n* = 5 per group, ***P* < 0.0001) in livers of AAV-GFP (GFP) and AAV-Cre (Cre) mice at D24 post virus injections. Scale bar: 50 µm. Unpaired *t*-tests, two-tailed in (**a**, **b**). **c** Representative Western blot in liver extracts of GFP and Cre mice at D21 and D24 post-virus injections (*n* = 4 per group). **d** Relative mRNA levels of mature hepatocyte markers and BEC/ progenitor-specific markers in livers as determined by qRT-PCR (*n* = 5). Multiple unpaired *t*-tests with the Holm-Sidak method. **P* < 0.05. **e** Representative immunofluorescence staining for HNF4a and KRT19 in liver sections. Scale bar: 50–100 µm. CV, central vein; PV, portal vein. **f** Representative immunofluorescence staining for Ki67 and Albumin (ALB). The numbers of Ki67⁺ (***P* < 0.0001) and Ki67⁺/ALB⁺ (**P* = 0.049) cells were quantified and normalized to the total numbers of DAPI-stained nuclei, respectively (*n* = 3 per group). Unpaired *t*-tests, two-tailed. All experiments in (**d**–**f**) were performed in the livers of male GFP and Cre mice at D24 post-virus injections. **P* < 0.05, ***P* < 0.005. All quantitative data were presented as mean ± SEM. Source data are provided as a Source Data file.

activation of fetal liver reprogramming. Immunofluorescence staining verified the reduction of HNF4α expression in most nuclear cells of AAV-Cre livers. Conversely, KRT19⁺ cells were more abundantly present in the periportal areas (Fig. 3e). Furthermore, the number of proliferating Ki67⁺ cells was significantly higher in AAV-Cre livers. This was evidenced by the presence of more Albumin (ALB) and Ki67 double-positive cells in AAV-Cre livers, suggesting increased hepatocyte regeneration (Fig. 3f). Consistently, LKO livers also displayed enhanced necrosis (Supplemental Fig. S3b), apoptosis (Supplemental Fig. S3c), and elevated expression of necroptosis- and apoptosis-associated proteins (Supplemental Fig. S3d). Transmission electron

microscopy (TEM) further identified ruptured membranes in hepatocytes of P10 LKO livers, indicative of early signs of necroptosis (Supplemental Fig. S3e). The livers from P14 LKO mice were also almost devoid of mature hepatocyte marker genes but enriched with BEC/ progenitor genes (Supplemental Fig. S3f). Double immunofluorescence staining revealed the specific reduction of ALB colocalized neddylation staining in hepatocytes but a marked increase in Pan-CK and NEDD8 double-positive cells in P14 LKO livers. Meanwhile, there were more activated hepatic stellate cells (HSCs) expressing desmin and NEDD8 in P14 LKO livers, whereas no significant differences in the staining of F4/80, a liver macrophage (Kupffer cell)

marker, and its colocalized NEDD8 were observed between two genotypes (Supplemental Fig. S4). Together, these data suggest that hepatic deletion of NAE1 in adult and neonatal mice lead to the initiation of apoptotic and necroptotic cell death concomitant with abnormal activation of fetal liver reprogramming and regeneration. The presence of more NEDD8-positive non-hepatocytes, such as BEC/progenitor cells and HSCs, likely contributes to the recovery of total neddylation observed in P14 LKO livers (Fig. 1a).

### Hepatic neddylation deficiency activates cell stress pathways with elevated oxidative stress and mitochondrial dysfunction

To unravel the primary molecular events affected by neddylation deficiency, we performed RNA-Seq to identify gene expression changes in early time points of P10 LKO and D21 AAV-Cre mouse livers. A total of 779 upregulated genes and 513 downregulated genes were identified in P10 LKO livers; whereas 2495 were upregulated and 1770 were downregulated in D21 AAV-Cre livers compared to their respective counterparts. Among all the differentially expressed genes (DEGs, $Log_2FC \geq 1.0$, $P_{adj} < 0.05$); 424 were upregulated and 232 were downregulated in both LKO and AAV-Cre livers (Fig. 4a). KEGG enrichment analysis of common DEGs identified top upregulated pathways involving glutathione (GSH) metabolism, drug metabolism–cytochrome P450, ErbB signaling, and TNF signaling; whereas metabolic pathways, specifically nitrogen metabolism, insulin signaling, bile secretion, and PPAR signaling were among those that were greatly suppressed (Fig. 4b). Comparing DEGs in D21 AAV-Cre mice showed positive correlations with expression changes of two ALF GSE38941 and GSE96851 transcriptomics, respectively (Fig. 4c). Heatmap revealed elevated expression of oxidative stress genes (e.g. *Cyp1a2, Cyp2a4, Gclc, Gclm, Gss, Gsr, and Mgst3*) and downregulated expression of metabolic genes (e.g. *Pck1, Ppara, Pgc1α, Srebp1, and Hmgcs2*) in both P10 LKO and D21 AAV-Cre livers (Supplemental Fig. S5a). Most of the DEGs related to oxidative stress and metabolic genes in AAV-Cre livers were dysregulated in both ALF GSE datasets (Supplemental Fig. S5b). The changes in those oxidative and metabolic genes were further validated by qRT-PCR in D21 AAV-Cre livers (Supplemental Fig. S5c). Indeed, there was increased reactive oxygen species (ROS) amounts in D21 AAV-Cre livers (Fig. 4d). Such changes were also evident in P10 LKO livers compared to WT livers (Supplemental Fig. S5d). When assessing the expression of various redox enzymes, we found a reduction of mitochondrial SOD1 and SOD2, but not Catalase, in D21 AAV-Cre livers. Interestingly, xanthine oxidase (XO) and NADPH oxidase 2 (NOX2), the major sources of cytoplasmic ROS mainly expressed by immune cells, were dramatically upregulated in D24 AAV-Cre livers (Supplemental Fig. S5e). Glutathione metabolic genes are well-known targets of NF-E2–related factor 2 (NRF2), an anti-oxidant transcription factor activated in response to oxidative stress[27]. Neddylation is known to activate CUL3-KEAP1-E3 ubiquitin ligase, which directly regulates NRF2 ubiquitination and protein degradation[28]. We found NRF2 expression was upregulated as early as D14 with a greater elevation by D21 in AAV-Cre livers (Supplemental Fig. S5e). Despite the dominant upregulation of GSH metabolism, the GSH levels were lower, whereas the GSSG levels, as well as the GSSG/GSH ratios, were higher in D21 AAV-Cre compared to AAV-GFP livers (Fig. 4e). These data suggest hepatic NAE1 deletion causes mitochondrial ROS ensued with GSH imbalance and exacerbated oxidative stress secondary to immune cell infiltration in vivo.

To identify whether oxidative stress is a downstream effect caused by NAE1 ablation, we isolated primary hepatocytes from D12 AAV-Cre livers and confirmed its deletion of NAE1 and reduced neddylation (Fig. 4f). Cultured NAE1-deleted primary hepatocytes exhibited elevated MitoSOX staining (Fig. 4g) and reduced GSH levels (Fig. 4h). Concurrently, these cells displayed suppressed basal, coupled, and uncoupled respiration (Fig. 4i, j) and ultimately decreased ATP contents (Fig. 4k). There was increased apoptosis (Fig. 4f) and reduced cell survival (Fig. 4l) in NAE1-deleted primary hepatocytes. In

line with in vivo findings, NAE1-deleted hepatocytes also demonstrated increased mRNA expression of oxidative stress genes and reduced metabolic genes (Supplemental Fig. S5f). Furthermore, MLN treatment of WT primary hepatocytes also caused higher mitochondrial ROS production (Supplemental Fig. S6a) and GSH depletion (Supplemental Fig. S6b), concurrent with a tendency of lower basal respiration but significantly downregulated maximal respiratory capacity (Supplemental Fig. S6c). While MLN alone did not affect cell survival and cause intrinsic apoptosis, it rendered primary hepatocytes more susceptible to exogenous TNFα-induced apoptosis (Supplemental Fig. S6d, e). Collectively, these data suggest that deficiency of NAE1-dependent neddylation triggers mitochondrial dysfunction, oxidative stress, and GSH deficiency, contributing to cell death.

### Hepatic neddylation deficiency autonomously activates mature hepatocyte fetal reprogramming

To identify whether hepatocyte reprogramming also occurred in vitro when neddylation is inhibited, we deleted NAE1 in *Nae1^{f/f}* primary hepatocytes via Cre adenoviral (pAd-Cre) infection. Cre virus transduction achieved ≈50% NAE1 knockdown and suppression of neddylation in *Nae1^{f/f}* primary hepatocytes. This resulted in the repression of mature hepatic markers (C/EBPα and HNF4a) at mRNA and protein levels and activation of BEC/progenitor cell gene expression (*Krt19* and *Opn*) (Fig. 5a). When WT primary hepatocytes were treated with MLN, similar upregulation of BEC/progenitor cell markers (*Krt19* and *Pkm*) was observed despite its resistance to MLN in downregulating hepatic markers (Supplemental Fig. S7a). We applied CRISPR/Cas9 knockout strategy and produced two distinct bulk cultures (KO1 and KO2) of HepG2 cells which exhibited >80% reductions in NAE1 protein expression and NEDD8 conjugation (e.g. neddylated CUL2). NAE1-deficient HepG2 cells exhibited a marked inhibition in mRNA and/or protein expression of C/EBPα and HNF4α, but a significant elevation in mRNA expression of *KRT19, OPN*, and *PKM* (Fig. 5b). Immunofluorescence staining of KRT19 further confirmed its upregulation in NAE1-deficient HepG2 cells (Fig. 5c). In neddylation-inhibited HepG2 cells, we identified similar hepatocyte reprogramming (Fig. 5d). These data suggest that neddylation deficiency induces hepatocyte reprogramming which is more robust in protumoral HepG2 cells.

Neddylation has been shown to modulate Hippo/YAP pathway activity[29], and constitutive activation of YAP is sufficient to promote hepatocytes to de-differentiate into stem-like cells[30]. We discovered that YAP protein expression was higher in MLN-treated primary hepatocytes, NAE1-deficient, or MLN-treated HepG2 cells (Supplemental Fig. S7a, b). However, YAP inhibitor Verteporfin neither suppressed the upregulation of BEC/progenitor cell markers nor restored HNF4α and C/EBPα expression, despite its ability to reduce *AMOLT2*, a known target of YAP, in MLN-treated HepG2 cells (Supplemental Fig. S7c, d). Furthermore, there was no consistent upregulation of YAP in NAE1-deleted livers in vivo (Supplemental Fig. S7e). Thus, reprogramming in neddylation-deficient hepatocytes is likely YAP-independent.

### Hepatic neddylation deficiency accumulates NIK and induces a non-canonical NF-κB pathway contributing to hepatocyte damage

Mice with hepatic overexpression of NIK demonstrate massive liver inflammation, oxidative stress, hepatocyte death, and liver fibrosis[21], recapitulating multiple aspects of the liver phenotypes observed in our hepatic NAE1-deficient mice. We next examined whether NIK is involved in neddylation deficiency-induced ALF. While endogenous NIK protein was not detected at basal conditions, it significantly accumulated in AAV-Cre livers, Ad-Cre-infected primary *Nae1^{f/f}* hepatocytes, NAE1-deficient HepG2 cells, and MLN-treated HepG2 cells (Fig. 6a, b, Supplemental Fig. S8a, b). Deletion of NAE1 in primary hepatocytes or neddylation inhibition in HepG2 cells did not affect the

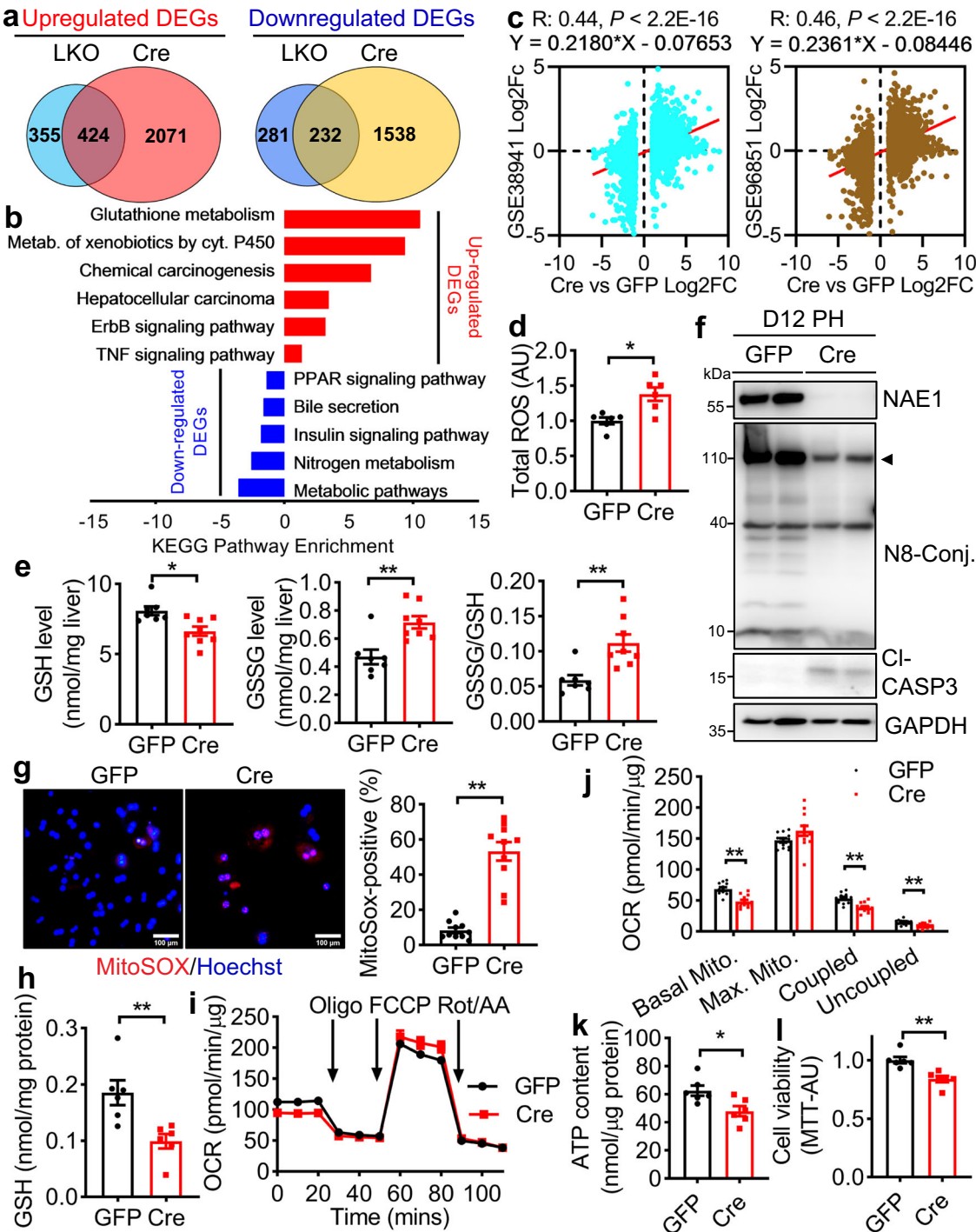

**Fig. 4 | Hepatocyte-specific neddylation deficiency induces oxidative stress and suppresses metabolic function. a** Venn diagrams showing comparisons of numbers of differentially expressed genes (DEGs, $1.0 \leq \text{LogFc} \leq -1.0$; *Pad* < 0.05) in livers from male P10 *Nae1^{f/f}*; *Alb-Cre* + (LKO) and D21 AAV-Cre (Cre) mice as compared to *Nae1^{f/f}* (WT) and AAV-GFP (GFP) counterparts, respectively. The intersection of the two circles shows DEGs in both LKO and Cre livers. **b** KEGG pathway enrichment analysis of DEGs significantly up-and downregulated in both LKO and Cre livers (*n* = 4 per group). **c** Scatter plots comparing DEGs of D21 Cre vs GFP with DEGs of GSE38941 (*t* = 27.89, df = 3304, 95% confidence intervals of 0.41–0.46) or GSE96851 (*t* = 29.56, df = 3304, 95% confidence intervals of 0.43–0.48), respectively. Cor.test() function in R was used. **d** Total ROS levels were measured by incubating liver homogenates with a DCFH-DA probe and expressed as AU (*n* = 6 per group, *P* = 0.0053). **e** GSH (*P* = 0.0075) and GSSG (**P* = 0.0031) levels normalized to liver weights and GSSG/GSH ratio (**P* = 0.0035) (GFP: *n* = 7; Cre: *n* = 8). Male D21 GFP and Cre mice and two-tailed unpaired *t*-tests were used in (**d**, **e**). **f–l** Primary hepatocytes (PH) were isolated from the livers of D12 male GFP and Cre mice. **f** Western blot (*n* = 2 per group); **g** MitoSOX staining and quantification of MitoSOX-positive cells from 10 images. Scale bar: 100 μm. **P* < 0.0001. **h** Total GSH levels normalized to protein levels (*n* = 6 per group). *P* = 0.0072. **i** Oxygen consumption rate (OCR) was detected using a Seahorse XF Cell Mito Stress Test. Basal, maximal (Max.), coupled, and uncoupled mitochondrial (Mito.) respirations were shown in **j**. **P* < 0.005. **k** Intracellular ATP contents normalized to protein levels (*n* = 6 per group). *P* = 0.0171. **l** Cell viability was determined by MTT assays and expressed as AU (*n* = 6 per group). **P* = 0.0018. Representative data from three biologically independent experiments were shown. Two-tailed Unpaired *t*-tests in (**g, h, k, l**). Multiple unpaired *t*-tests with the Holm-Sidak method in (**j**). GFP (black), Cre (red). All quantitative data were presented as mean ± SEM. Source data are provided as a Source Data file.

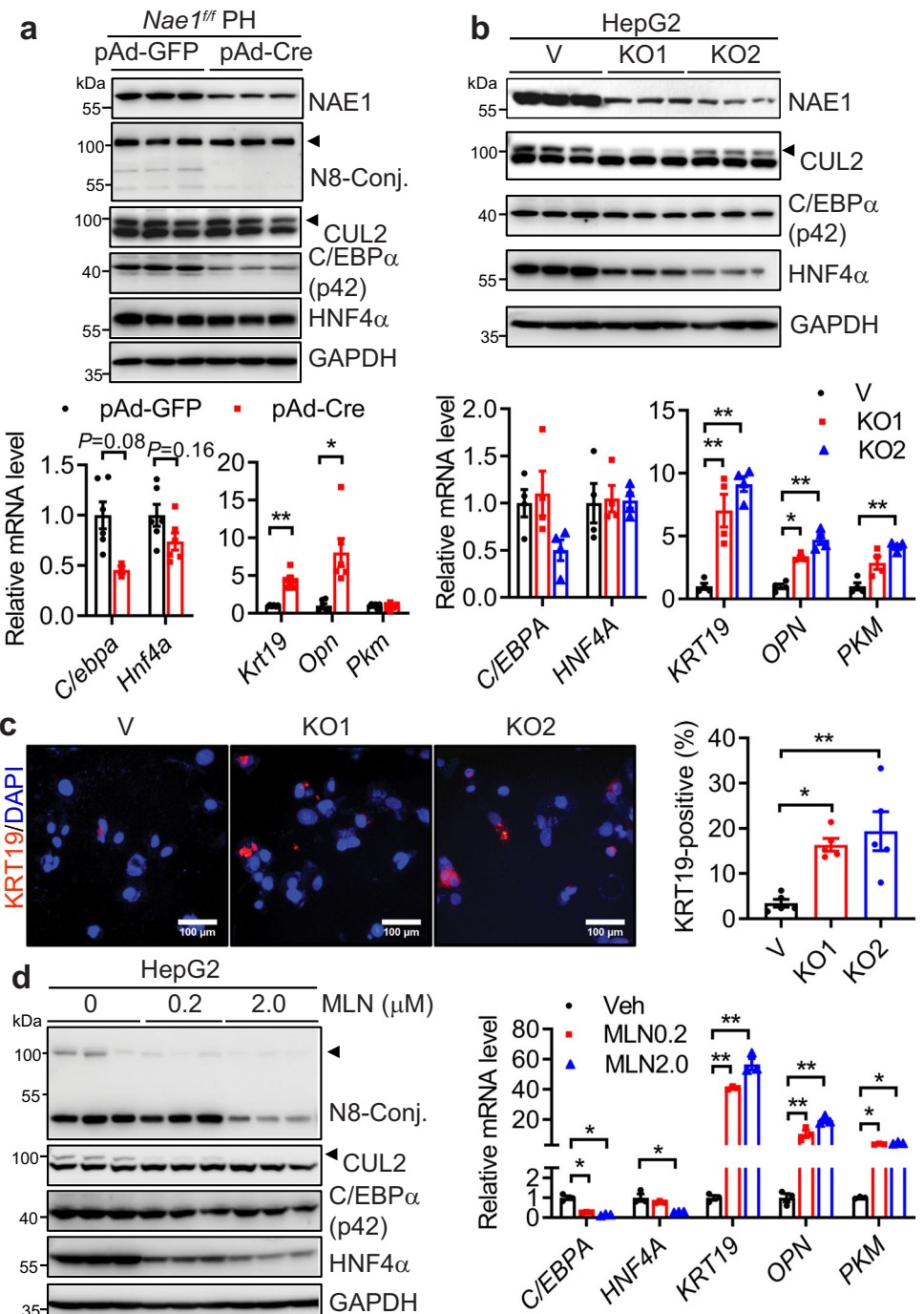

**Fig. 5 | Neddylation deficiency autonomously activates hepatocyte fetal reprogramming. a** Representative Western blot and relative mRNA levels of mature hepatocyte and BEC/ progenitor-specific marker genes in mouse *Nae1^f/f* primary hepatocytes (PH) infected with pAd-GFP and pAd-Cre for 48 h (*n* = 6 per group). pAd-GFP (black), pAd-Cre (red). Multiple unpaired *t*-tests with the Holm-Sidak method. **b** HepG2 cells were infected with pLentiCRISPR/Cas9 V2 lentiviruses bearing two specific human NAE1 sgRNAs to generate bulk cultures with NAE1 knockout (KO1 and KO2), respectively. Cells infected with viruses expressing no sgRNA were used as control (V). Western blot and relative mRNA levels of mature hepatocyte and BEC/progenitor-specific marker genes were analyzed (*n* = 4 per

group). V (black), KO1 (red), KO2 (blue). Two-way ANOVA followed by Tukey's multiple comparisons test. **c** Representative immunofluorescence staining for KRT19 and quantification of KRT19⁺ cells (*n* = 5 per group). Scale bar: 100 µm. V (black), KO1 (red), KO2 (blue). One-way ANOVA with Tukey's multiple comparisons test. **d** Representative Western blot and relative mRNA levels of mature hepatocyte and progenitor/BEC-specific marker genes in HepG2 cells treated with vehicle (Veh) or indicated concentrations of MLN for 48 h (*n* = 3 per group). Veh (black), MLN0.2 (red), MLN2.0 (blue). Two-way ANOVA followed by Tukey's multiple comparisons test. *$P < 0.05$, **$P < 0.005$. All quantitative data were presented as mean ± SEM. Source data are provided as a Source Data file.

mRNA levels of NIK (Supplemental Fig. S8c). While the proteasome inhibitor bortezomib (BZM) or MLN increased endogenous NIK expression, co-treatment of BZM and MLN synergistically elevated NIK protein accumulation (Supplemental Fig. S8d). To determine whether neddylation directly regulates NIK stability, we performed

cycloheximide (CHX)-based pulse-chase assay to assess the degradation of ectopically expressed NIK. Our results revealed that MLN significantly prolonged the half-life the of NIK protein (Fig. 6c). These in vivo and in vitro data indicate that neddylation promotes the degradation of NIK in a cell-autonomous manner.

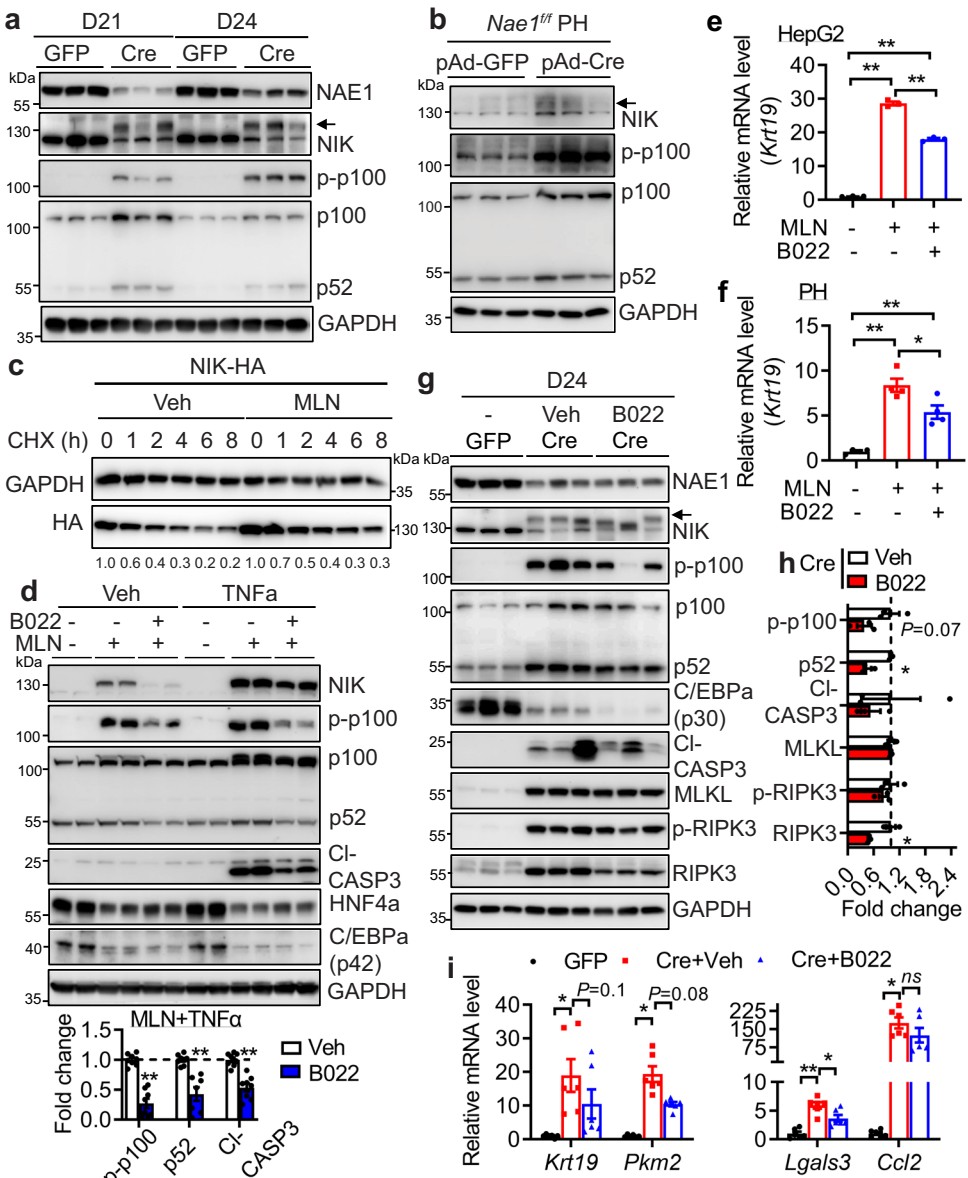

**Fig. 6 | Neddylation deficiency elevates NIK which contributes to hepatocyte damage. a** Western blot in whole liver extracts of AAV-GFP (GFP) and AAV-Cre (Cre) mice at D21 and D24 post-virus injections (*n* = 6 per group). **b** Immunoblot analyses in mouse *Nae1^f/f* primary hepatocytes (PH) 48 h after infection with pAd-GFP and pAd-Cre (*n* = 3 per group). Arrow indicates the expected murine NIK. **c** Eighteen hours after transfection with NIK-HA, HepG2 cells were treated with vehicle (Veh) or 1.0 μM MLN for 12 h before 20 μg/ml CHX treatment for indicated periods. **d** Western blot and quantifications in HepG2 cells treated with Veh, 1.0 μM MLN with or without 5 μM B022 for 48 h. 20 ng/mL TNFα was added in the last 24 h. Cleaved caspase-3 and NIK signaling were quantified by normalizing to cells receiving MLN and TNFα (*n* = 5 per group). Veh (black), B022 (blue). Multiple unpaired *t*-tests with the Holm-Sidak method. **e, f** qRT-PCR analyses in (**e**) HepG2 cells (*n* = 3 per group) and (**f**) wild-type primary hepatocytes (*n* = 4 per group)

treated with vehicle (Veh), 1.0 μM MLN with or without 5 μM B022 for 48 h. In **e, f**, Veh (black), MLN (red), MLN + B022 (blue). One-way ANOVA followed by Tukey's multiple comparisons test. **g–i** Male *Nae1^f/f* mice injected with AAV-Cre received Veh or B022 via daily *i.p* injection from D10 post-virus injection. AAV-GFP mice were also included as a control. **g, h** Western blot of indicated proteins and quantifications in liver extracts from D24 AAV-GFP, AAV-Cre mice receiving Veh or B022 (*n* = 3 per group). Data in **h** were normalized to AAV-Cre mice receiving Veh. Multiple unpaired *t*-tests with the Holm-Sidak method. Veh (black), B022 (red). **i** Relative mRNA levels in livers from D24 AAV-GFP, AAV-Cre receiving Veh or B022 mice (GFP: black, *n* = 6; Cre+Veh: red, *n* = 6; Cre+B022: blue, *n* = 3 in duplicates). Two-way ANOVA followed by Tukey's multiple comparisons test. *P < 0.05; **P < 0.005; ns: not significant. All quantitative data were presented as mean ± SEM. Source data are provided as a Source Data file.

NIK is essential to activate the non-canonical NF-κB pathway[20]. Indeed, accumulation of NIK proteins in NAE1-deficient livers was accompanied by an increase in phosphorylated p100 at Ser866/870, total p100, and its cleaved product p52, the classic biomarker for NIK activation of non-canonical NF-κB pathway (Fig. 6a). The increase in p100 could be due to impaired CRL1-mediated p100 ubiquitination and degradation[31] secondary to neddylation deficiency. We confirmed increased p52 in the nuclear fractions of both D21 and D24 AAV-Cre livers (Supplemental Fig. S8e), further supporting the activation of the

non-canonical NF-κB pathway. Notably, increases in phosphorylated p100 and p52 were also observed in NAE1-deficient primary hepatocytes, NAE1-deleted, or MLN-treated HepG2 cells (Fig. 6b, Supplemental Fig. S8a, b). In contrast to the activation of the non-canonical NF-κB pathway in NAE1-deficient cells, inhibition of neddylation had little effect on p65 expression under all tested conditions (Supplemental Figs. S8a, e–g). Instead, neddylation inhibition significantly increased phosphorylated IκBα with a negligible effect on total IκBα protein levels (Supplemental Figs. S8a, S8f–g). These data suggest that

neddylation deficiency causes a more consistent activation of NIK-mediated non-canonical NF-κB signaling, but not IκBα-mediated canonical NF-κB signaling in hepatocytes.

To investigate the functional relevance of NIK activation to neddylation deficiency-induced hepatocyte damage, we applied a specific NIK inhibitor B022 to suppress NIK activity and its downstream signaling[32]. B022 significantly attenuated MLN-induced aberrant NIK signaling (Fig. 6d) and ameliorated *KRT19* upregulation in MLN-treated HepG2 cells or primary hepatocytes (Fig. 6e-f). NAE1 deletion is highly associated with upregulated hepatic TNF signaling (Fig. 4b). TNFα accentuated MLN-induced NIK accumulation. Although B022 failed to improve the expression of hepatocyte marker genes (*HNF4A*, *C/EBPA)*, key GSH regulatory gene (*GCLC*), and gluconeogenic gene (*PCK1*) in basal or TNFα treated cells, MLN-induced sensitization to TNFα-triggered apoptosis could be mitigated by B022 treatment (Fig. 6d and Supplemental Fig. S8h). These data suggest NIK activation contributes to neddylation deficiency-triggered hepatocyte damage.

In vivo administration of B022 to mice for 14 days led to a 50% reduction of phosphorylated p100 and p52 accumulation in AAV-Cre livers (Fig. 6g-h), demonstrating its effectiveness in blocking NIK signaling. While B022 had little effect on the caspase-3 cleavage in AAV-Cre livers, it significantly mitigated RIPK3 upregulation without dampening its phosphorylation and MLKL expression as compared to Veh-treated AAV-Cre livers (Fig. 6g-h). While neither hepatocyte marker genes nor oxidative and fibrotic genes were affected by B022 treatment in AAV-Cre mice (Supplemental Fig. S8i), B022 significantly attenuated the expression of inflammatory genes (e.g., *Lgals3*, but not *Ccl2*) and showed a trend, though not statistically significant, toward suppressing the activation of progenitor genes (*Krt19*, *Pkm*) in AAV-Cre livers (Fig. 6i). Despite these beneficial effects, the employed B022 regimen was not sufficient to rescue the mortality of AAV-Cre mice (Supplemental Fig. S8j). Likewise, the administration of apoptosis and necroptosis inhibitors (Z-VAD/NEC1) to AAV-Cre mice also failed to rescue its lethality (Supplemental Fig. S8j). Infliximab, a monoclonal antibody that blocks TNF signaling exhibited a trend toward attenuating the lethal phenotype in AAV-Cre mice (Supplemental Fig. S8j). These data suggest that targeting any single perturbed pathway in the NAE1-deleted liver is not sufficient to rescue neddylation deficiency-induced ALF in vivo.

## NIK is a direct neddylation substrate and neddylated NIK promotes its degradation

Given the notorious role of NIK in a broad spectrum of liver diseases including ALF[21, 22], we next investigated how neddylation regulates the stability of NIK. We first examined whether neddylation-activated CRLs mediate NIK expression. Deletion of CUL1, CUL2, CUL3, or CUL4a in either HepG2 or NIH-3T3 cells by CRISPR/Cas9 did not alter NIK expression (Fig. S9a-d). Deletion of the catalytic subunit RING-box 2 (RBX2), which pairs with CUL5 to form CRL5[33, 34], did not change NIK expression in HepG2 cells either (Fig. S9e). Thus, these data do not support the direct role of CRLs in the control of NIK degradation.

We next determined whether NIK is a direct neddylation substrate by performing denaturing immunoprecipitation of ectopically expressed NIK in HepG2 cells overexpressing wild-type (WT) NEDD8 or its conjugation-deficient mutant (ΔGG). Western blot of immunoprecipitates detected smearing neddylated NIK species in WT-NEDD8-expressing cells but not in NEDD8$^{\Delta GG}$-expressing cells (Fig. 7a). Inhibition of neddylation by MLN treatment or silencing neddylation E2 enzyme UBC12 both robustly reduced the abundance of neddylated NIK (Figs. 7a-7b, Supplemental Fig. S9f). Meanwhile, overexpression of wild-type deneddylase SENP8[35], but not the catalytically inactive C163A mutant, robustly reduced the levels of neddylated NIK (Fig. 7c). In contrast, deletion or downregulation of SENP8 resulted in the accumulation of total neddylated proteins and increased neddylated NIK (Fig. 7d). Immunoprecipitation further identified an interaction

between endogenous UBC12 and endogenous NIK in HepG2 cells and primary hepatocytes, supporting the involvement of the neddylation enzyme in NIK neddylation (Fig. 7e). Collectively, these data identify NIK as a bona fide neddylation substrate.

Human NIK contains 59 lysine residues, and 49 are conserved between mouse and human NIK (Fig. 7f). We next sought to identify potential neddylation sites on NIK. Mutation of all 49 conserved lysine residues (K49R) or the C-terminal (aa 385-947) 20 conserved lysine residues (K20R) abrogated NIK neddylation, whereas mutation of the N-terminal (aa 1-384) 29 conserved lysine residues (K29R) did not alter NIK neddylation status (Fig. 7g), suggesting that the C-terminal 20 conserved lysines serve as the major NEDD8 acceptor sites. Tryptic digestion of neddylated proteins produces neddylated peptides with a unique K-ε-GG signature that can be identified by mass spectrometry[36]. Mass spectrometry analysis of NIK immunoprecipitates identified 4 lysines (K385/387/607/650) carrying di-glycine motif among C-terminal 20 lysines, which were only detectable in cells co-overexpressing NIK and NEDD8 (Supplemental Fig. S10a, b). Mutation of any single lysine site (K385R, K387R, K607R, and K650R) at these four lysines had minimal impact on NIK neddylation (Supplementary Fig. S10c), however, mutation of all four lysine residues (K385R/K387R/K607R/K650R; K4R) significantly decreased the levels of neddylated NIK (Fig. 7h), suggesting that all four lysine residues are putative NEDD8 accepting sites.

We next explored whether NIK neddylation affects its ubiquitination and degradation. Depletion of UBC12 or NEDD8 significantly reduced NIK ubiquitination after denaturing pulldown assays (Fig. 7i). NIK ubiquitination was also diminished after MLN treatment, suggesting neddylation of NIK positively regulates its ubiquitination (Supplemental Fig. S10d). Interestingly, NIK K4R appeared to be more stable than its WT counterpart, because the protein level of NIK K4R was always higher than that of its WT counterpart when the same amount of plasmid was transfected (Fig. 7j). However, such differences were abrogated upon BZM or MLN treatment, suggesting these four sites are involved in mediating neddylation- or proteasome-regulated NIK protein stability (Fig. 7j). Indeed, NIK K4R mutant had a slower turnover rate than that of WT (Fig. 7k). Thus, we propose that neddylation of NIK promotes its ubiquitination and degradation.

## NAC therapy partially rescues neddylation deficiency-induced ALF

Neddylation is well known to target E3 ubiquitin ligases such as CRLs to mediate protein ubiquitination and degradation, and ubiquitin pathways regulate the pathogenesis of various liver diseases[37]. Interestingly, the total ubiquitination levels were upregulated in AAV-Cre livers. Whether such changes are due to the inactivation of deubiquitinating enzymes or alterations in the cellular population remains unknown. Nevertheless, in addition to NRF2 and IκBα (Fig. S5e and Fig. S8), we found two additional CRL-regulated proteins: p53 and CTNNB1 (i.e., β-Catenin)[37, 38] were significantly elevated in AAV-Cre livers. Aberrant expression of p53[39] or CTNNB1[40] is known to regulate liver metabolic dysfunction, oxidative stress, and inflammation. Therefore, neddylation deficiency also disrupts ubiquitination pathways to contribute to ALF (Supplemental Fig. S11a).

Since neddylation deficiency perturbs various signaling pathways that converge to oxidative stress and inflammation, we next turned to NAC, a GSH surrogate and antioxidant that is effective in treating ALF and disease-associated oxidative stress and inflammation[41, 42]. In HepG2 cells, NAC marginally suppressed MLN-induced *KRT19* upregulation without restoring the expression of hepatocyte marker genes (*HNF4A*, *C/EBPA)*. However, it successfully reverted GCLC upregulation (Fig. 8a) and GSH deficiency (Supplemental Fig. S11b). It also recovered *PCK1* gene expression and protein expression of C/EBPα, indicative of restored gluconeogenic capacity (Fig. 8a, b). NAC was more effective in suppressing MLN-induced *Krt19* upregulation and replenishing *Pck1*

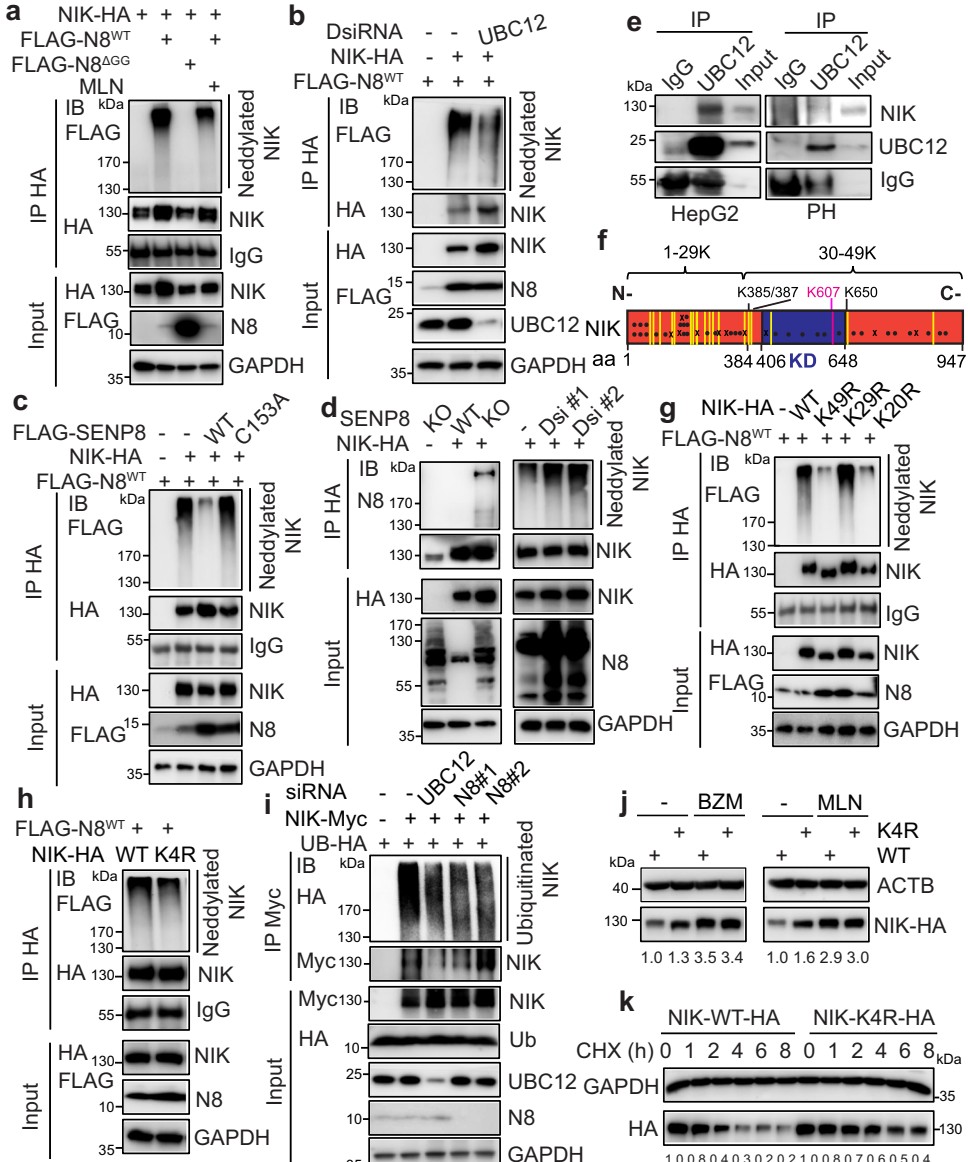

**Fig. 7 | NIK is neddylated and neddylation promotes its ubiquitination and degradation. a** Twenty-four hours after transfection with the indicated constructs, HepG2 cells were treated with or without 2.0 μM MLN for 24 h. The neddylation of NIK-HA was examined by IB after denaturing IP. Cell lysates were IB as input. N8^WT, wild-type NEDD8; N8^ΔGG, conjugation-deficient NEDD8 mutant. **b** The neddylation of NIK-HA was examined by IB after denaturing IP in 293 T cells transfected with the indicated constructs with scrambled DsiRNA or DsiRNA against UBC12. **c, d** NIK neddylation was assessed after denaturing IP in (**c**) 293 T cells overexpressing SENP8 WT and C153A mutant, (**d**) WT and SENP8-KO MEFs, and 293 T cells with SENP8 knocked down. Cells were co-transfected with indicated NIK-HA and/or FLAG-N8^WT. **e** Western blot after IP with an anti-UBC12 antibody or a control IgG antibody in HepG2 and primary hepatocyte (PH) lysates treated with BZM (100 nM, 12 h). **f** A schematic diagram of the lysine sites on NIK is shown. ·: conserved, x: non-conserved lysines. Yellow lines indicate lysine residues previously identified with K-ε-GG remnants. Catalytic kinase domain (KD): 405-646 amino acids (aa).

**g** Neddylation of various NIK K to R mutants examined by IB after denaturing IP in HepG2 cells. **h** Proteomic analyses identified the four potential neddylation sites (K385, K387, K650 indicated by yellow lines, and K607 in purple). Forty-eight hours after transfection in HepG2 cells, the neddylation of NIK-HA (WT) and the corresponding 4 R mutant (all of four lysine residues mutated to arginine) was then examined by IB after denaturing IP. **i** NIK ubiquitination was analyzed after denaturing IP in 293 T cells transfected with the indicated constructs or (D)siRNAs against UBC12 or NEDD8. BZM was added 8 h before harvesting. **j, k** Twenty-four hours after transfection with the same (**j**) or adjusted (**k**) amounts of the indicated plasmids, HepG2 cells were treated with BZM (100 nM, 8 h) or MLN (1.0 μM, 24 h) or 20 μg/ml CHX for indicated periods followed by Western blot. Quantifications were shown as numbers under each image. Three biologically independent experiments except for mass spectroscopic analysis. Source data are provided as a Source Data file.

expression in primary hepatocytes (Supplemental Fig. S11c). Meanwhile, co-treating MLN with NAC substantially diminished hepatocyte sensitivity to TNFα-induced apoptosis (Fig. 8b). These data suggest that NAC is effective in repairing neddylation deficiency-induced hepatocyte damage.

We next tested whether NAC administration in drinking water ameliorates neddylation deficiency-induced ALF in vivo. When D10

AAV-Cre mice were fed with NAC water, we identified NAC feeding significantly increased GSH contents (Supplemental Fig. S11d) and reduced hepatic ROS levels (Supplemental Fig. S11e) in AAV-Cre mice to the same levels as NAC-fed AAV-GFP mice. As expected, the mortality in NAC-fed AAV-Cre mice decreased significantly to 70% versus 100% in AAV-Cre mice receiving regular water (Fig. 8c). While the plasma ALT levels only trended lower, the plasma bilirubin levels in

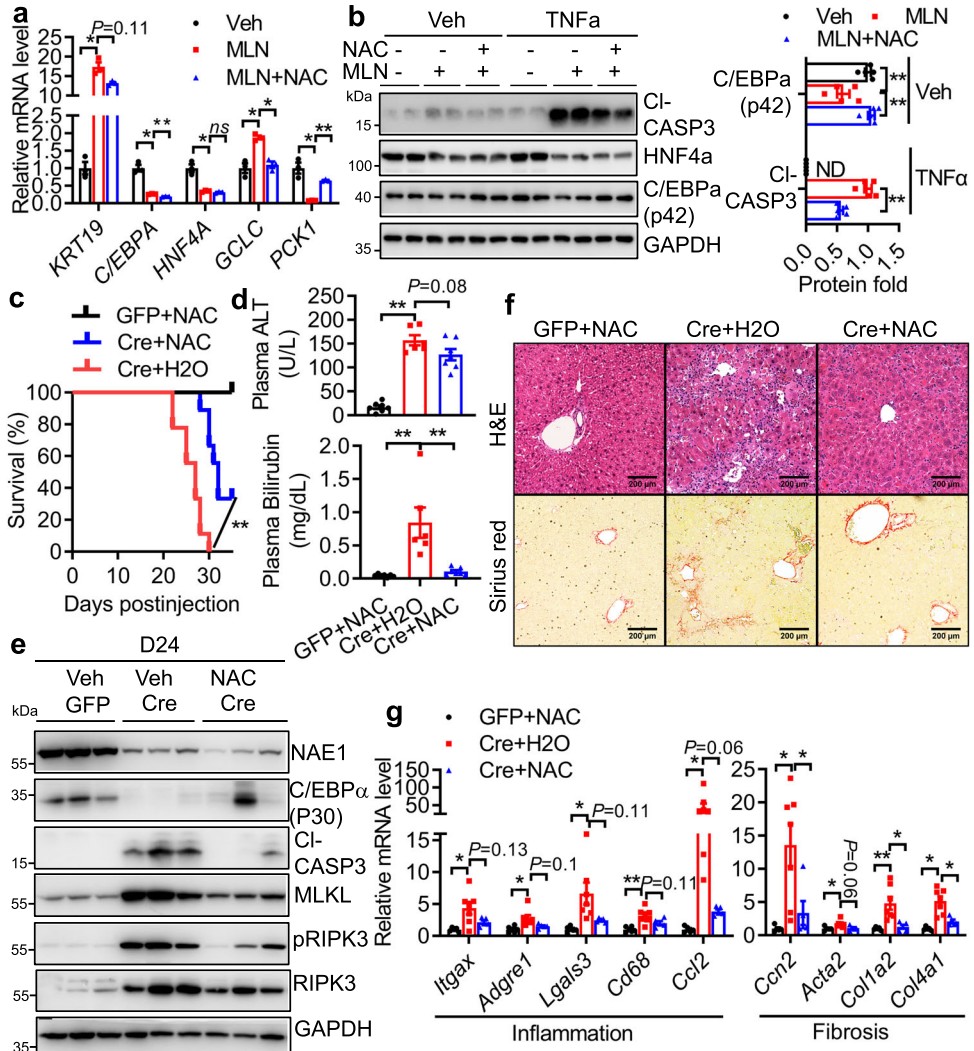

**Fig. 8 | NAC ameliorates NAE1 deletion-induced lethal liver failure. a** qRT-PCR analyses in HepG2 cells treated with vehicle (Veh), 1.0 μM MLN with or without 10 mM NAC for 48 h (*n* = 3 per group). **b** Representative Western blot and quantifications in HepG2 cells treated with Veh, 1.0 μM MLN with or without 10 mM NAC for 48 h. 20 ng/mL TNFα was added in the last 24 h (*n* = 5). C/EBPα protein expression was normalized to cells receiving Veh. Cleaved caspase-3 activation was normalized to cells receiving TNFα. Three biologically independent experiments. In (**a**, **b**), Veh (black), MLN (blue), MLN + NAC (blue). Two-way ANOVA followed by Tukey's multiple comparisons test. \**P* < 0.05; \*\**P* < 0.005. ns: not significant. ND: not detected. **c**–**g** Male 10-12-week-old *Nae1^{f/f}* mice injected with AAV-Cre received 1% NAC-supplemented drinking water (Cre+NAC) or regular water (Cre+H₂O) from D10 post-virus injection. AAV-GFP mice receiving NAC water (GFP + NAC) served as

a control. **c** Survival curves (GFP + NAC: *n* = 5; Cre+H2O: *n* = 9; Cre+NAC: *n* = 9). Log-rank (Mantel-Cox) test. \*\**P* = 0.0002. **d** Plasma ALT and bilirubin levels (GFP + NAC: *n* = 7; Cre+H2O: *n* = 6; Cre+NAC: *n* = 7). One-way ANOVA followed by Tukey's multiple comparisons test. \*\**P* < 0.005. **e** Representative Western blot of indicated proteins in liver extracts (*n* = 5 per group). **f** Representative H&E and Sirius red staining of liver sections (*n* = 3 per group). Scale bar: 200 μm. **g** Relative mRNA levels of inflammatory and fibrotic genes (GFP + NAC: *n* = 5; Cre+H2O: *n* = 7; Cre +NAC: *n* = 5). Two-way ANOVA followed by Tukey's multiple comparisons test. \**P* < 0.05; \*\**P* < 0.005. **d**–**g** were performed in mice at D24 post-virus injection. GFP + NAC (black), Cre+H₂O (red), and Cre+NAC (blue). All quantitative data were presented as mean ± SEM. Source data are provided as a Source Data file.

NAC-fed AAV-Cre mice were completely restored to the normal levels as NAC-fed AAV-GFP mice (Fig. 8d). Meanwhile, oral NAC feeding in AAV-Cre mice markedly attenuated hypoglycemia (Supplemental Fig. S11f) and slightly ameliorated hypercholesterolemia (Supplemental Fig. S11g). While NAC feeding did not affect NAE1 deletion in the AAV-Cre group, it moderately recovered the mRNA and protein expression of hepatocyte markers and suppressed the upregulation of BEC/progenitor marker genes in the NAC-fed AAV-Cre mice (Fig. 8e and Supplemental Fig. S11h). Data from immunoblotting further revealed weakened upregulation of apoptotic and necroptotic proteins in the livers of AAV-Cre mice treated with NAC (Fig. 8e). Consistently, NAC feeding improved liver architecture and reduced inflammatory cell infiltration (Fig. 8f). The improvement of liver injury was further verified by a tendency of lower expression of inflammatory genes and significantly diminished expression of cytokines in NAC-fed

AAV-Cre livers (Fig. 8g). Such changes were also associated with decreased fibrosis as supported by weakened Sirius red staining (Fig. 8f and Supplemental Fig. S11i) and fibrotic gene expression (Fig. 8g) in NAC-fed AAV-Cre livers compared with regular water-fed AAV-Cre livers. Collectively, these data suggest that NAC therapy could partially protect mice from NAE1 deletion-induced hepatocyte death, inflammation, fibrosis, and thus lethal hepatic failure.

## Discussion

In this study, we found that hepatic NAE1 deficiency induces extensive liver damage consistent with ALF seen clinically. We provided compelling evidence that hepatocyte neddylation is important in maintaining postnatal and adult liver homeostasis, and a proper level of hepatocyte neddylation is critical for maintaining normal liver function. Mechanistically, our data support the multiple-hit hypothesis

which suggests neddylation deficiency simultaneously disrupts divergent signaling pathways, including but not limited to mitochondrial dysfunction, oxidative stress, hepatocyte reprogramming, and NIK activation. Given the power of neddylation in the liver, simply blocking one single pathway (NIK, TNF, or cell death) may not be sufficient to rescue the fatal phenotype seen in hepatic NAE1-deficient mice. Our findings further demonstrate that NAC is effective in treating ALF in AAV-Cre mice, similar to its efficacy in human ALF.

Disruption of redox and GSH homeostasis has been well associated with the ALF of various etiologies[43]. Our data support the notion that hepatic neddylation deficiency consistently induces oxidative damage and reduces intracellular GSH levels, contributing to inflammation, fibrosis, and liver failure. However, genes for GSH biosynthesis were markedly increased in response to neddylation inhibition, indicating an indirect role of neddylation deficiency in causing GSH deficiency. Notably, neddylation deficiency upregulated NRF2-mediated antioxidant pathway. However, the induction of NRF2 and its targeted GSH metabolic genes was insufficient to be hepatoprotective, suggesting the antioxidant capacities were overwhelmed by ROS production from dysfunctional mitochondria and inflammation. NAC displays both direct and indirect antioxidant properties and is more effective in treating diseases associated with oxidative stress and inflammation[41, 42, 44]. The successful amelioration of hepatocyte death and liver failure by GSH-replenishing NAC suggests oxidative stress might be involved in ALF caused by hepatic NAE1 deficiency. Notably, NAC was effective only when given at the early phase of NAE1 deletion (i.e., at D10). One limitation of our study is that we only performed adult-onset deletion of NAE1 in hepatocytes and NAC treatment in male mice. Whether similar findings occur in females is not known. Nevertheless, our studies provide evidence of the efficacy of NAC in neddylation deficiency-induced ALF.

Mitochondrial dysfunction plays an important role in liver lesions by exerting deleterious consequences, such as oxidative stress, energy shortage, steatosis, and cell death[45]. Hepatic steatosis is more obvious in adult NAE1-deleted livers, which may result from a more severe mitochondrial dysfunction. Both NAE1 deletion and inhibition in hepatocytes caused mitochondrial ROS and respiratory dysfunction. The mitochondrial dysfunction in cultured NAE1-deleted primary hepatocytes was probably underestimated given the selective attachment of relatively viable hepatocytes. MLN has been shown to suppress basal but not maximal oxidative phosphorylation in protumoral hepatocytes[9]. In our hands, murine primary hepatocytes were relatively tolerant to MLN which induced mitochondrial ROS but only blocked maximal oxidative phosphorylation. The exact mechanisms underlying NAE1 deletion-induced mitochondrial dysfunction remain elusive. A previous report identified that mitochondrial proteins are directly neddylated[16]. Specifically, neddylation directly targets and stabilizes electron transfer flavoproteins (ETFs). Inhibition of neddylation reduces the protein levels of ETFs, leading to suppressed fatty acid oxidation[16]. Undoubtedly, more neddylation-targeted mitochondrial proteins remain to be identified. Conversely, MLN has been shown to stimulate hepatocyte mitochondrial function and fatty acid oxidation to ameliorate steatosis in mouse models of dietary nonalcoholic fatty liver disease (NAFLD)[46]. The protective role of MLN was only observed under disease states with higher neddylation levels. Our current evidence suggests that neddylation, although relatively low in healthy adult hepatocytes[47], is required for the maintenance of liver homeostasis under physiological conditions. Therefore, its level needs to be carefully fine-tuned when applying neddylation inhibition-based therapies to avoid potential adverse effects on liver function.

While neddylation deficiency in hepatocytes might facilitate both intrinsic and extrinsic caspase-mediated apoptosis, it is unlikely to trigger hepatic necroptosis, as we were not able to observe RIPK3 and MLKL upregulation in cultured neddylation-deficient hepatocytes. Only in NAE1-deleted livers could we identify necroptosis, suggesting

extrinsic inflammatory pathways contribute to NAE1 deletion-induced hepatocyte destruction by necroptosis. The mechanisms and pathogenesis of cell death from neddylation deficiency may be multifactorial due to the complex nature of neddylation-modified targets and pathways.

The levels of progenitor cell accumulation correlate positively with acute mortality in human ALF[48]. While hepatocytes are known to undergo reprogramming in response to massive liver damage, our data suggest that neddylation deficiency may autonomously reprogram adult hepatocytes to acquire a fetal-like phenotype. Whether the neddylation-deficient hepatocytes are fully reprogrammed to the progenitors or simply present as hybrid hepatocytes for liver regeneration in vivo remains unknown. Notch and Hippo/YAP signaling pathways have been shown to mediate the critical switch between BECs and hepatocyte specification[30, 49]. In our case, we found YAP upregulation in neddylation-deficient hepatocytes. However, this pathway does not seem to be the culprit, as inhibiting YAP signaling failed to repress hepatocyte reprogramming. Notably, mitochondrial dysfunction and oxidative stress have been shown to trigger a niche favoring BEC/progenitor cell expansion as an adaptive response[50]. Indeed, NAC treatment suppressed abnormal fetal reprogramming in AAV-Cre livers. While both LKO and AAV-Cre livers demonstrated fetal liver reprogramming, the ductular reaction was more overt in LKO livers, which presented the unique formation of cyst-like structures. Notably, Alb-Cre achieves efficient gene deletion in both BECs and hepatocytes at the perinatal state, a critical time point for intrahepatic bile duct development[51]. Thus, it is plausible that neddylation deficiency in BECs may directly perturb bile duct development postnatally, leading to cyst formation in LKO livers. The exact mechanisms underlying hepatocyte reprogramming and cyst formation warrant further in-depth studies. Nevertheless, considering the primary role of hepatocytes in the post-injury liver is self-renewal[52], our data suggests neddylation deficiency causes excessive fetal liver reprogramming, causatively potentiating ALF.

Metabolic stress, oxidative stress, and cytokines stimulate hepatic NIK[22]. While these factors may contribute to NIK activation in NAE1-deleted liver, our study uncovers a direct role of neddylation in mediating NIK expression. We provided a series of data to support that NIK is a genuine substrate of neddylation. (i) NEDD8 is covalently attached to NIK. (ii) NIK neddylation depends on an activating enzyme (E1, NAE1) and a conjugating enzyme (E2, UBC12), and can be inhibited by MLN, a specific E1 inhibitor. (iii) SENP8 is a specific deneddylase of NIK. (iv) Endogenous NIK associates with endogenous UBC12. (v) 20 conserved lysines within 385-947 amino acids are the major NEDD8 acceptor sites in NIK, including four putative neddylation sites identified by our proteomics (K385/387/607/650). Notably, neddylation in the NIK K4R mutant was not completely abolished when these 4 sites were mutated, suggesting a potential shift of modification sites or the presence of additional neddylation sites. NIK ubiquitination sites are largely different from its neddylation sites, as 16 out of 21 ubiquitination sites are localized at the N-terminus of NIK (Fig. 7f, yellow line)[53–55]. However, we cannot rule out the possibility that the identified k-ε-GG peptides are modified by ubiquitin since both ubiquitination and neddylation generate the same K-ε-GG remnants after trypsin digestion. The general limitation of proteomic analysis, together with the presence of numerous lysines in NIK preclude us from clearly demarcating the neddylation and ubiquitination sites. SMURF2[56], JUNB[57], and c-Src[58] can be neddylated and degraded in a ubiquitin-dependent manner. Here, we also propose that NIK neddylation promotes its ubiquitination and proteasome-mediated degradation, since an impairment of neddylation by knockdown of UBC12 or NEDD8 or treatment of MLN suppresses NIK ubiquitination, and NIK K4R mutant exhibits improved protein stability. It is possible that neddylation alters NIK conformation to expose its ubiquitination sites, and/or improve the

binding affinity of NIK to its ubiquitin ligases to promote its ubiquitination. Such a possibility remains to be tested in the future.

Aberrant activation of NIK is reported to trigger fatal liver inflammation independent of its downstream non-canonical NF-kB pathway[21]; block hepatocyte reparative replication to impair liver regeneration[59], and impede fatty acid oxidation by restraining PPARα activity thus promoting alcoholic steatosis in mice[60]. While these pathways may be involved in neddylation deficiency-induced ALF progression, we further identified that NIK signaling blockade by B022 could ameliorate KRT19 upregulation and the synergistic cytotoxicity of TNFα and MLN in cultured hepatocytes. Such findings were further corroborated by the alleviation of progenitor activation and inflammation in B022-treated AAV-Cre mice. However, it remains to be determined whether a more potent inhibition of NIK in vivo would result in a significant improvement in mortality in AAV-Cre mice. Nonetheless, our data suggest that neddylation deficiency causes inappropriate activation of NIK, contributing to the pathogenesis of hepatic failure. Such regulation may have broad implications for the clinical effects of MLN on various NIK-linked diseases.

Interestingly, hepatic deletion of NAE1 (our data), UBA3, or NEDD8 mediated by Alb-Cre all led to neonatal death[16]. However, cysts were only present in our *Nae1^{LKO}* mice. While induced deletion of NAE1 in adult mature hepatocytes recapitulates ALF, mice with hepatic loss of UBA3 and NEDD8 in adult mice failed to show any difference in body and liver weights even after a 6-month period[16]. Notably, we used a more hepatocyte-specific AAV8-TBG-Cre to delete NAE1, whereas AAV-DJ Cre recombinase driven by a strong synthetic CAG promoter was used in Zhang et al. to induce genetic deletion of UBA3 or NEDD8 potentially in all liver cells[16]. Whether such differences resulted in the stark phenotypic differences between these adult mutant mice is not known. It is worthwhile to note that NAE1 has been shown to interact with UBE1C[61] and TRIP12[62]. Thus, it remains interesting to determine whether NAE1 mediates neddylation-independent functions to regulate liver development and function, differently from UBA3 and NEDD8.

In summary, our study uncovered an indispensable role of neddylation in regulating the function and fate of postnatal and adult hepatocytes. Additionally, we established a model of neddylation deficiency-induced ALF and identified the downstream targets of neddylation which may endow us with insights into therapeutically intervening ALF progression.

## Methods

### Animals and treatment
All animal experiments were conducted according to the NIH guidelines for the care and use of laboratory animals and approved by the IACUC at Augusta University (approval #: 2012-0462). All transgenic mice used were in the C57BL/6 J background. Hepatocyte-specific *Nae1* knockout mice (*Nae1^{LKO}*) were generated by crossing *Nae1* floxed (*Nae1^{f/f}*) mice[29] with *Alb-Cre* transgenic mice (JAX#: 003574). For in vivo viral transduction to specifically delete NAE1 in mature hepatocytes, 12-week-old *Nae1^{f/f}* mice were injected via tail vein with adeno-associated viral vector serotype 8 (AAV8) bearing a hepatocyte-specific thyroxine-binding globulin (TBG) promoter-driven eGFP (AAV8-TBG-eGFP, VB1743, Vector Bio-lab) or Cre (AAV8-TBG-Cre, VB1724, Vector Bio-lab) expression at $1.0 \times 10^{11}$ genome copies per mouse. A dose of 1% NAC (Sigma-Aldrich) was added to the drinking water[63]. The NIK inhibitor B022 (MedChemExpress, HY-120501), Necrostatin 1 (NEC1, Cayman Chemicals, 11658), and Z-VAD-FMK (Z-VAD, MedChemExpress, HY-16658B) were dissolved in 10% DMSO, 40% PEG300, 5% Tween-80 and 45% saline. B022 was daily intraperitoneally (*i.p.*) administered at a dose of 25 mg/kg/day[60]. NEC1 and Z-VAD were daily co-administered at doses of 1.65 mg/kg/d[64] and 6 mg/kg/d[65], respectively. Infliximab (Y0002047, Sigma-Aldrich) was reconstituted in saline and *i.p.* administered at a dose of 10 mg/kg every other day[66].

All treatments were started in mice 10 days post-virus injections. Other than the scientific requirement to analyze mortality, animals with hepatic deletion of NAE1 were monitored daily and sacrificed before they manifested 20% body weight loss. Mice were not fasted before sacrifice by cervical dislocation. Tissues were fixed or snap-frozen at −80 °C. Animals were kept under a controlled temperature of around 21 °C and 50% humidity with a 12 h/12 h light and dark cycle.

### Measurement of plasma parameters and liver hydroxyproline content
Non-fasting blood glucose levels were measured using a glucose meter (One Touch Ultra, Life Scan). Total plasma levels of TG, cholesterol, ALT, and bilirubin were measured in colorimetric analyses using Infinity Triglycerides (ThermoFisher Scientific, TR2241) and Infinity Cholesterol Kits (ThermoFisher Scientific, TR-13421), ALT colorimetric activity assay kit (Cayman Chemical, 700260) and a total bilirubin reagent set (Pointe Scientific, 23-666-150) according to the manufacturer's instructions, respectively. Hydroxyproline assay to quantify collagen content was performed using Hydroxyproline Assay Kit (Cell Biolabs, Inc, STA-675) according to the manufacturer's instruction. Data were normalized to liver weights.

### Histology, Sirius red staining, TUNEL, and EBD labeling, and immunostaining
Liver tissues were fixed in 10% formalin, paraffin-embedded, sectioned, and stained with H&E. Pathological scoring was performed blindly by a board-certified liver pathologist based on the drug-induced liver injury (DILI) scoring system[67]. Liver fibrosis was assessed using 0.1% Sirius Red stains (Sigma-Aldrich) for 1 h followed by two washes of acidified water (0.5% acetic acid) in liver sections as described[68]. Hepatic apoptosis was determined by **TUNEL** staining using an In Situ Cell Death Detection Kit (Sigma-Aldrich, 11684795910) according to the manufacturer's instructions. **EBD labeling** was used to identify necrotic hepatocytes on a dye-exclusion basis as previously described[69]. Briefly, mice were anesthetized and *i.p.* injected with 1% EBD (w/v, Sigma-Aldrich) at 10 ml/kg BW. 24 h post-injection, the mice were anesthetized and perfused with PBS followed by 4% PFA. Harvested liver tissues were sucrose-dehydrated and then OCT-embedded. For **immunostaining**, OCT-embedded liver frozen sections were permeabilized with 0.2% Triton X-100. Cells were fixed with 4% paraformaldehyde and permeabilized with 0.02% Triton X-100. Tissue sections or cells were blocked with 5% normal goat serum, incubated with primary antibodies followed by corresponding fluorescence-conjugated secondary antibodies (Invitrogen). All slides were counterstained and mounted with Prolong Gold antifade mountant with DAPI (Invitrogen), and then visualized under an All-in-One fluorescence microscope (Keyence). The percentages of stained areas or positive cells were quantified from eight randomly selected fields per section using two sections of each mouse and three mice per genotype by ImageJ 1.52t software (NIH).

### RNA sequencing and bioinformatics analysis
RNA-seq was performed at the Genome Technology Access Center at Washington University. Total RNA was extracted by Trizol (Invitrogen) from liver tissues of P10 WT and LKO mice as well as D21 AAV-GFP and AAV-Cre mice post virus injection (male, 4 h fast, $n = 4$ with each pooled from 3 animals). The library was prepared using a Ribo-Zero gold rRNA removal kit and paired-end sequencing of 50 bases length was performed on a HiSeq 3000 system (Illumina). The RNA-Seq basecalls files were converted to fastq files by using the bcl2fastq (version 2.15.0.4) program. RNA-seq reads were then aligned to the Ensembl release 76 top-level assemblies with STAR version 2.0.4b. Gene counts were derived from the number of uniquely aligned unambiguous reads by Subread: feature count version 1.4.5. Quantification, normalization, and DEGs were determined with the Cufflinks package (version 2.2.1).

DEGs were used as an input for pathway analysis through Ingenuity Pathway Analysis (IPA) suite (www.ingenuity.com, accessed 2019). The R/Bioconductor package heatmap3 was used to display heatmaps or annotated KEGG graphs across groups of samples for each KEGG pathway with a Benjamini-Hochberg false-discovery rate adjusted P-value less than or equal to 0.05. DEGs were defined as ones with at least 0.5 FPKM level of expression in at least 1 of the conditions and a Q-value <0.05.

## Seahorse bioenergetics

Primary hepatocytes were isolated and plated at $2 \times 10^4$ cells per well into Seahorse XF24 tissue culture plates. Oxygen consumption rate (OCR) was measured using the Seahorse Mito Stress test kit (Agilent) as instructed. OCR was normalized to total protein levels.

## Immunoblotting and immunoprecipitation

Proteins from tissues or cells were extracted in a lysis buffer (50 mM Tris, pH 7.5, 150 mM NaCl, 1.5 mM $MgCl_2$, 1 mM EDTA, 1% Triton X-100, 10% glycerol) or RIPA buffer (ThermoFisher Scientific) freshly supplemented with protease and phosphatase inhibitor cocktails (Sigma-Aldrich). Protein concentrations were quantified by Bradford protein assay (Bio-Rad) or Pierce™ BCA Protein Assay Kit (ThermoFisher Scientific). For non-denaturing immunoprecipitation, cells were lysed in the above lysis buffer, and lysates were incubated with the respective antibody-conjugated beads for 16 h at 4 °C. Precipitates were washed three times and eluted. For denaturing immunoprecipitation, cells were lysed in a denaturing buffer (1% SDS, 5 mM EDTA, 10 mM DTT), boiled at 95 °C for 5 min, and then diluted 10 times with the 1% Triton X-100 lysis buffer before immunoprecipitation. Equal amounts of total protein lysates or immunoprecipitates were analyzed by standard Western blotting procedures. Protein bands were developed using the ECL substrate (Millipore) and visualized by Amersham Imager followed by densitometry qualification using ImageQuantTL v8.2.0 (GE Healthcare). A description of the antibodies is provided in Supplementary Table 1.

Uncropped blots are available in the Source data file and the Supplementary information.

## Quantitative Real-time PCR (qRT-PCR)

Total RNAs were extracted from liver tissues or cells using TRIzol reagent (Invitrogen) according to the manufacturer's protocol. The first-strand cDNAs were synthesized using random primers and M-MLV reverse transcriptase (Invitrogen). qRT-PCR was performed with SYBR Green Supermix (Bio-Rad) using the Stratagene Mx3005p system (Agilent Technologies). Relative mRNA abundance was calculated after normalization to the housekeeping genes (*Ppia* and *36B4*) based on the Genorm algorithm (medgen.ugent.be/genorm/) and expressed as fold changes. All qRT-PCR primers were obtained from Integrated DNA Technology. The sequences of primers are listed in Supplementary Table 2.

## Cell Culture, transient transfection, lentiviral and adenoviral generation, and generation of NAE1-deficient HepG2 cells using CRISPR/Cas9

The HepG2 (HB-8065), HEK-293 (CRL-1573), HEK-293T (CRL-3216), and NIH/3T3 (CRL-1658) cell lines were obtained from ATCC and cultured in DMEM (Sigma-Aldrich) supplemented with 10% fetal bovine serum (FBS) and Penicillin-Streptomycin. Primary hepatocytes were isolated from 3-month-old male mice by a modified in situ collagenase perfusion procedure as described previously[70]. Briefly, the inferior vena cava is cannulated and the liver is perfused with 5 ml liver perfusion media (ThermoFisher Scientific) followed by 8–10 ml perfusion of 0.1% collagenase IV (Sigma). Hepatocytes were then dissociated by spatula, filtered through a 100-μM filter, and spun down at 100xg for 3 mins followed by one additional spin after resuspension in Percoll solution.

After the final wash, cells were plated on the collagen-coated plates in DMEM supplemented with 10% FBS and Penicillin-Streptomycin. Four to six hours after plating, cells were changed to William E media supplemented with antibiotics and 2% FBS followed by virus infection or treatment. SENP8 WT and KO mouse embryonic fibroblasts (MEFs) were prepared from E14 embryos derived from SENP8 WT and KO mice (kindly provided by Dr. Huabo Su). All cells were cultured at 37 °C in a humidified atmosphere containing 5% $CO_2$. Cells were transiently transfected with plasmids using PolyJet™ in vitro DNA transfection reagent or X-tremeGENE HP DNA Transfection Reagent, siRNA, or Dicer-Substrate Short Interfering RNAs (DsiRNAs) using GenMute™ siRNA transfection reagent (SignaGen Laboratories, Frederick, MD) according to the manufacturer's instructions.

HEK 293 T cells were plated in a 6-well plate and transfected with the guide RNA plasmid (pLenti-CRISPR V2 empty vector or vector encoding sgRNAs against human NAE1, RBX1, CULs GenScript), pMD2.G-VSV-G, and psPAX2 using PolyJet™ in vitro DNA transfection reagent for 6 h before changing to complete cell culture media. 48 h after transfection, the supernatant containing the lentivirus was collected and centrifuged at 10,000 x g for 10 min. HepG2 or NIH/3T3 cells were plated in a 6-well plate at $5 \times 10^5$ cells per well. 24 h after plating, cells were changed to media containing fresh media and lentivirus supernatant (1:1) in the presence of 8 μg/mL polybrene overnight. The media was changed to fresh media containing 2.0 μg/ml puromycin for at least 5 days. Cells were harvested for the determination of deletion by immunoblotting.

Adenovirus vectors bearing CMV promoter-driven expression of GFP (pAd-GFP) and Cre (pAd-Cre) were amplified in HEK293 cells, purified, and titrated according to protocol[71]. Primary hepatocytes were infected with pAd-GFP and pAd-Cre at 1:500 MOI. As this vector contains an independent CMV-driven transcription cassette, the efficiency of transduction can be directly monitored by visualization of the GFP expression.

The following drugs were used in cell culture studies: Cycloheximide and bortezomib (Sigma-Aldrich), B022 (MedChemExpress), MLN4924 (MilliporeSigma), Verteporfin (Cayman Chemicals), mouse and human recombinant TNF-A (R@D Diagnostics Ltd). All other drugs were obtained from Sigma-Aldrich. All the plasmids, sgRNA sequences (GenScript), DsiRNA (IDT), or siRNA (Sigma) sequences were provided in Supplementary Tables 3 and 4, respectively.

## Statistics and reproducibility

All the animal experiments were carried out in at least two independent cohorts, except RNA-Seq and survival analyses. All quantitative data were presented as mean ± SEM. At least two biologically independent experiments were performed for cellular studies except for proteomic analysis. Statistical analysis was performed with GraphPad Prism 9.1.2 with either two-tailed unpaired t-test, multiple t-tests after correction using the Holm-Sidak method, or one-way ANOVA corrected for Tukey's multiple comparisons test, or two-way ANOVA followed by Tukey's posthoc tests as dictated by each experiment. Survival curves were analyzed by Log-rank (Mantel-Cox) test. Cor.test() function in R was used to compute correlation coefficient. A P-value of <0.05 was considered statistically significant.

## Reporting summary

Further information on research design is available in the Nature Portfolio Reporting Summary linked to this article.

## Data availability

The RNA-Seq data generated in this study have been submitted to the SRA database with the BioProject ID: PRJNA693817. The previously published gene expression data from patients with hepatitis B virus-associated acute liver failure re-analyzed in this study were obtained through the Gene Expression Omnibus, GSE38941[23] and GSE96851[24].

The mass spectrometry proteomics data have been deposited to the ProteomeXchange Consortium via the PRIDE[72] partner repository with the dataset identifier PXD038653. All other data generated or analyzed during this study are included in this published article (and its supplementary information files). Source data are provided in this paper. Source data are provided with this paper.

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

## Acknowledgements

We thank the Genome Technology Access Center (partially supported by NCI Support Grant #P30 CA91842 ICTS/CTSA Grant# UL1TR002345 from the National Center for Research Resources (NCRR) in the Department of Genetics at Washington University School of Medicine for help with genomic analysis). This work was supported by National Heart, Lung, and Blood Institute [R01HL132182 to W.C., R01HL124248 to H.S.], and the American Heart Association Career Development Award (18CDA34080244 to H.Z.).

## Author contributions

C.X. and W.C. designed the research, conducted experiments, acquired and analyzed data, and wrote the manuscript. H.Z., A.R., and K.S. performed experiments and data analysis; Y.J. and A.B. performed liver pathological scoring and discussed the results. Y.L. and K.D. analyzed human and mouse liver RNA-seq data. H.S. and J.Z. provided technical and material support, discussed the results, and edited the manuscript. W.C. supervised the work.

## Competing interests
The authors declare no competing interests.
