## [Peer Review File · Nature Communications]

Hepatic neddylation deficiency triggers fatal liver injury via inducing NF- κ B-inducing kinase in miceREVIEWER COMMENTS

Reviewer #1 (Remarks to the Author):

The authors show that perinatal or adult impairment of Neddylation causes liver failure. The derangements are very broad and challenge the interpretation of which are cause or effect. The Neddylation effect on NIK expression is novel but the contribution to ALF is unclear.

1 – Although the KEGG analysis modestly point to TNF signaling (4B), the expression of TNF was even less striking (1H and 2F). The only strong support for importance is the HepG2 experiment which shows sensitization to TNF but many hepatic insults are known to sensitize to TNF. The effect of TNF blocker (e.g. infliximab) or receptor knockdown in vivo should be examined.

2 – Evidence of apoptosis and necroptosis is presented. Each could be triggered by TNF but not likely in the same cells at the same time. Need IHC to assess which cells are involved and if there are zonal differences in apoptosis or necrosis.

3 – The impact of pancaspase inhibitor (e.g. Z-VAD) and RIPK1 inhibitor (Nec-1s) or both should be assessed to determine which cell death mechanism accounts for the liver failure.

4 - The authors show many pathologic and biochemical effects of inhibiting Neddylation. There is obviously a profound disturbance of hepatic metabolism making it difficult to pin-point the key events, though oxidative stress appears to account for some.

5 – The mechanism of oxidative stress certainly may be linked to GSH depletion and mitochondrial impairment but cause and effect are not clearly defined. The mechanistic link between NIK and the biochemical pathologic events needs to be explored more deeply. As presented this is not directly elucidated.

Reviewer #2 (Remarks to the Author):

In this manuscript, the authors aim to clarify the role of neddylation deficiency in the regulation of acute liver injury. They reported that hepatic-specific inhibition of neddylation by NAE1 deletion caused hepatocyte damage, mitochondrial dysfunction, and excessive fetal liver reprogramming. Besides, the authors identified NIK as a novel NEDD8 target. The authors also reported that antioxidant therapy partially rescued NAE1 deficiency-induced ALF. Although the topic of the research and some of discoveries the authors presented are interesting, many of data are not solid enough to support the conclusions. Some of major issues that need to be addressed are listed below.

1. The pattern of global protein Neddylation in multiple figures is quite different (e.g. Figs 1A/2A/5A/6A/6B/6D). In Figure 2A, there is a band above 55kD, while in the figure 5A, there is a band detected in 40kD. When NAE1 is downregulated, the neddylation of CULs should be inhibited. Therefore, neddylation modification of cullins, as the classic substrates of Neddylation, should be detected upon NAE1 downregulation.

2. It is well known that neddylation deficiency causes mitochondrial dysfunction, apoptosis and cell death. The authors also showed that hepatocyte-specific neddylation deficiency caused hepatocyte damage, mitochondrial dysfunction, and excessive fetal liver reprogramming. Are the mechanisms of these phenotypes the same as the reported mechanisms of Neddylation inhibition, such as activating intrinsic and extrinsic apoptosis?
3. It is well known that the inactivation of Neddylation-CRL pathway inhibited the degradation of I κ B α , leading to the inactivation of NF- κ B pathway. The authors showed the regulation of non-canonical NF- κ B pathway through hepatocyte-specific neddylation deficiency. How about the canonical NF- κ B pathway?
4. In figure 6A, the authors claimed that they achieved \approx 50% NAE1 knockdown and suppression of neddylation in pAd-Cre. However, the N8-Conjugates had no obvious decrease upon NAE1 downregulation.
5. As mentioned by the authors, in figure 6B, in the NAE1-KO HepG2 cells, the N8-Conjugates strongly increased. The increase of N8-conjugates is quite confusing. The authors should detect the classic substrates of Neddylation (e.g. CUL1, CUL2) to further confirm the neddylation inhibition after knockdown of NAE1.
6. In figure 6B, NAE1 knockdown is more efficient in KO2 compared with KO1. However, in figure S7C, the expression of NIK and p-p100 in KO1 is stronger than that in KO2. Meanwhile, the accumulation of p52 is not obvious enough.
7. In figure 7, the authors showed that NAE1 knockdown induced NIK accumulation, and MLN4924 treatment delayed the degradation of NIK (Fig. 7A-B). Co-treatment MLN and BZM synergistically elevated endogenous NIK protein accumulation (Fig. 7C). Therefore, the authors claimed that neddylation deficiency stabilizes NIK protein in a proteasome-dependent manner. All these results strongly indicate NIK may be a substrate of CRL E3 ligase, but not a direct substrate of neddylation. The authors should determine whether NIK is a substrate of CRL E3 ligase.
8. In figure 7, the authors tried to prove that NIK is a neddylation substrate, however, the data provided are not sufficient to support this conclusion. Much more work is needed to strengthen the conclusion. Besides, in figure 7D, the Flag-NEDD8 band in input is very confusing. In figure 7E, the endogenous co-IP assay result is not satisfactory. I would say the interaction is so weak that it cannot be concluded that UBC12 interacted with NIK.
9. The authors claimed that MLN dose-dependently induced post-transcriptional NIK protein expression and signaling in HepG2 cells (Figure S7, D-E) (the last lane of page 11 and first line of page 12). However, in figure S7E, the mRNA level of Mapk3k14 also seem to be upregulated, so this conclusion is not very clear. The authors did not provide the corresponding significance analysis in the bar plot or legend.
10. In Figure S6, the figure legend is not corresponding to the figure.

11. In figure 3C, the C/EBP α is labeled P30. In figure 6A/6B/6D, the C/EBP α is labeled p42. What is the difference?

Reviewer #3 (Remarks to the Author):

NCOMMS-21-06607 Review

In this manuscript, the authors investigate a role for NAE1 and neddylation in human ALF disease. Through the use of conditional deletion of NAE1 in the embryo (Alb-CRE) and the adult knockouts, they show that reduction of neddylation leads to phenotypes similar to ALF, such as hepatocyte injury and death, liver inflammation and fibrosis. Gene Expression from both the embryonic knockout (LKO) and adult knockout show very similar responses in gene expression, as well as pathways that point increased oxidative stress and mitochondrial dysfunction which are thought to contribute to the liver phenotype.

They also show the decrease in mature hepatocyte gene expression and increase in genes and proliferation (Ki67+) reflecting BEC/progenitor following injury. This suggests a role for cellular reprogramming and/or regeneration in the pathophysiology of this phenotype. They also demonstrated a cell autonomous effect in CRISPR NAE1 knockouts in HepG2 cells and with neddylation inhibitor MLN in vitro with similar effects on gene expression.

Mechanistically, they show a connection between Neddylation and NF- κ B inducing kinase (NIK) stability. NIK is highly regulated by ubiquitination and degradation and is stabilized by a subset of TNF receptor family members upstream of alternative NF- κ B signaling. They show a similar stabilization and activation of NIK in the absence or inhibition of neddylation enzymes leading presumably to p100 phosphorylation and increased p52 processing. They also demonstrate a benefit with oxidative stress inhibition (NAC) on the mouse liver phenotype.

Comments:

(1) Only figure 6 refers to 3 independent experiments with representative images and plots of data. Would the authors please clarify how many times each set of observations was performed including embryonic and adult knockout cohorts?

(2) The authors suggest that there is significant perturbation of neddylation enzyme expression in two ALF patient datasets, with NAE1 being significantly downregulated. If they have access to ALF patient samples, measuring NAE1 expression for themselves would be preferable and would make a nice figure 1 A to set up the manuscript for why they have targeted NAE1. Authors may not have access to these human samples, but measuring NAE1 expression in multiple patient samples may raise the quality of the data and further support their reference to the phenotype in this disease as Figure 1A rather than in the

supplement. Seeing the spread of the data in the patient population compared with control would be important.

(3) In figure 1 A, the authors show a nice reduction in N8-Conj in the KO P7 and p10, but this effect goes away by P14. Would the authors please comment on why this might be. Are there compensatory mechanisms that occur over time? Do other neddylation enzymes also use N8 as a substrate?

(4) In figure 1, the liver injury, inflammation and fibrosis are clear. Curious whether they looked at hydroxyproline levels and/or collagen gene expression in the embryonic knockout? Collagen genes were measured in the adult knockout.

(5) In reference to figure 1F, they refer to plasma ALT as a measure of liver function. While that is correct for plasma bilirubin, plasma ALT is a measure of hepatocyte damage and death not function.

(6) The quality of the plot in figure 1H is poor. It is hard to see the dynamic range of the effect when the symbols are so low against the axis. This plot should be redone so that we can effectively see the fold change in expression of these genes.

(7) Would the authors speculate on why they observe steatosis in the adult but not in the embryonic knockout?

(8) In figure 3D, the reduction of mature hepatocyte markers and increase in BEC/progenitor markers is interesting. Is this due to mature hepatocyte death, reprogramming of some cells to progenitors or expansion of existing progenitors in the liver? It is hard to tell whether this effect is due to cellularity differences in the bulk tissue sample. Since this is RNA from bulk tissue, have the authors thought about doing single cell RNAseq to better tease out the cell populations that are present in these mice? The Ki67+ staining does suggest increased progenitor activation following injury, but given the current analysis it is hard to tell what they are and where they are coming from especially since the increase in progenitor activation (if it is occurring) does not keep up with the hepatocyte damage in this model. Is this an example of “frustrated” progenitors that are not capable of further differentiation following activation? Single cell RNAseq may better define known progenitor populations in the liver and give a more complete data set.

(9) Figure 4 shows a nice correlation between the embryonic and adult knockouts. Have the authors correlated these genes with the ALF data sets to see how relevant their model is to human AFL disease beyond just NAE1 expression and other neddylation enzymes? This would strengthen the relevance of the paper to human disease.

(10) Figure 5 F Y-axis should start at 0 as not to overemphasize the effect between groups.

(11) Figure 6 A, the authors should consider plotting C/ebpa and Hnf4a differently as to be able to see the fold change between groups. Although referred to as significant it is too small a difference as plotted to see effectively.

(12) Regarding Figure 6 A and B, would the authors please comment on the relatively limited effect in reduction of mature hepatocyte markers C/ebpa and Hnf4a. The effect between the two appears different between primary hepatocytes and HepG2 cells. This is less dramatic than the gene expression in bulk tissue. The authors should comment on why that might be and if this truly represents “reprogramming” to a progenitor like state or just transient gene expression differences. Again, single cell might give a more holistic representation of what is going on here.

(13) The authors clearly show the effect of neddylation on NIK stability. What is lacking in this dataset are the downstream effects of increased NIK stability/activity. The effect is mostly done with

overexpressed NIK protein. Overexpression of NIK can have effects that may be different from endogenous levels of regulated NIK. Endogenous NIK can be hard to detect and pull down, but have they tried to look at the same effect on endogenous NIK? Have the authors thought about using proteomic approaches to observe neddylated and ubiquitinated NIK peptides?

(14) Figure 7A, the results appear inconsistent. Would the authors please comment on how many times these results were observed and how many individual mice were tested by western? The day 24 p52 blot is not very good. Is this whole tissue/cell lysate or nuclear extract? Nuclear extracts generally reflect a clearer increase in p52. Also curious what the total p100 bands look like above the p52 band. Do they observe a reduction in the p100 band given the rather modest effect on p52 in this case?

(15) In figure 7, MLN inhibition leading to reduced neddylated NIK is not very strong. I do acknowledge that the IP western detection is not very quantitative, however the reduction in figure 7D & E is not very strong. Did the authors titrate the amount of MLN used and test earlier timing of the MLN treatment post HepG2 transfection to get a more optimal result? Does the residual neddylation represent the amount of NIK neddylated prior to MLN treatment? Would the authors speculate on the rate of turnover of neddylated NIK.

(16) The authors make nice case for the effect of reduced oxidative stress (NAC treatment) on the NAE1 knockout phenotypes. Reduction in liver injury, inflammation, and fibrosis, gene expression as well as the mature hepatocyte vs BEC/progenitor genes, with clear controls to show reduction in markers of oxidative stress (supplement). What is lacking is the same rigor of working out the role of NIK in these phenotypes. Given the importance the authors are putting on the NIK neddylation connection, the same depth of investigation into the effect of NIK knockout or inhibition on the NAE1 / MLN phenotype in vivo would be important. The authors do point to the what has been shown in the literature about NIK transgene on liver injury / disease phenotypes, however, what is lacking is the direct effect of NIK inhibition / knockout on neddylation deficient phenotypes, such as hepatocyte injury and death, mature vs progenitor gene expression balance, oxidative stress, etc.

(17) Authors do show a 50% reduction in HepG2 apoptosis (Casp3) with NIK inhibitor B022 induced by TNF α similarly to NAC, but do not go beyond that. If the B022 compound is the one that I am familiar with from Amgen, the compound lacks significant potency as well as has selectivity limitations. Have the authors tried treating with B022 to see if NIK is required for the effect of NAE1 knockout or MLN on HepG2 induction of progenitor genes and repression of mature hepatocyte genes?

Overall, this manuscript demonstrates a clear effect of neddylation deficiency on liver injury and disease. The following should be addressed for acceptance:

(1) The authors show a reduction in mature hepatocyte markers and an increase in progenitor markers and suggest that there might be cellular reprogramming going on in response to liver injury. Rather than relying on bulk mRNA changes, the authors should use single cell RNAseq to identify cell populations that are present at the time of disease which would be greatly informative here.

(2) A correlation between knockout RNAseq data sets with ALF correlated genes would greatly strengthen their story and the proposed connection between NAE1/Neddylation and ALF liver disease.

(3) More correlation or efficacy data with NIK knockout or inhibition in the context of NAE1 knockout induced liver disease. While the NIK biochemical results on neddylation are intriguing, the proposed

hypothesis of the relationship to human disease requires more direct links between NIK activity and downstream alternative NF- κ B signaling with neddylation induced disease and possible relationship to human ALF would be important to see if accepted for Nature Communications.

RESPONSES TO REFEREES

We thank the Editor and Reviewers for the constructive comments. We have significantly improved the manuscript according to the helpful suggestions. We combined the original **Fig. 4** and **Fig. 5** to a new **Fig. 4**. We performed proteomics analysis and identified the neddylation sites of NIK. These sites were mutated to test their involvement in mediating NIK ubiquitination and protein stability. We further examined the contribution of NIK signaling to neddylation deficiency-induced acute liver failure (ALF) both *in vitro* and *in vivo* using the NIK specific inhibitor B022. In addition, we also investigated whether blocking TNF or apoptosis/necroptosis signaling pathways could ameliorate NAE1 deletion-induced ALF, respectively. These data are included into **new Fig. 6, Fig. 7** and **Supplemental Figs. S7 and S8**. Below is the point-by-point response to the comments.

Reviewer #1 (Remarks to the Author):

The authors show that perinatal or adult impairment of Neddylation causes liver failure. The derangements are very broad and challenge the interpretation of which are cause or effect. The Neddylation effect on NIK expression is novel but the contribution to ALF is unclear.

1 – Although the KEGG analysis modestly point to TNF signaling (4B), the expression of TNF was even less striking (1H and 2F). The only strong support for importance is the HepG2 experiment which shows sensitization to TNF but many hepatic insults are known to sensitize to TNF. The effect of TNF blocker (e.g. infliximab) or receptor knockdown *in vivo* should be examined.

Response: We agree with the reviewer. In both P14 LKO and D21 Cre livers, the upregulated genes involved in TNF signaling pathway include inflammatory cytokines (*Ccl2* and *Lif*), chemokines (*Cxcl10*), interferon gamma inducible protein 47 (*Irf47*), transcription factor (*Jun*), necroptosis marker (*Mkl1*) and adhesion molecule (*Sele*). These genes are all downstream of TNF signaling. We did not find that *Tnf* was significantly upregulated in neonatal NAE1-deleted liver at P14, a relatively early stage of disease progression (**Fig. 1h**), but it was elevated in D24 AAV-Cre livers (**Fig. 2f**). To directly address whether TNF signaling contributes to neddylation deficiency-induced liver injury, we treated D10 AAV-Cre mice with Infliximab (the TNF blocker) every other day and monitored their survival compared to vehicle-treated AAV-Cre mice. Infliximab only produced a trend toward improving the survival in NAE1 deletion-induced acute liver failure (Please see **Supplemental Fig. S8h**). These data suggest the potential involvement of TNF signaling in neddylation deficiency-induced ALF. However, divergent inflammatory pathways are linked to neddylation deficiency. Blocking TNF alone is not sufficient to reduce mortality.

2 – Evidence of apoptosis and necroptosis is presented. Each could be triggered by TNF but not likely in the same cells at the same time. Need IHC to assess which cells are involved and if there are zonal differences in apoptosis or necrosis.

Response: We agree with the reviewer. Our human liver pathologists carefully re-examined liver histology. Both P14 LKO and D24 AAV-Cre livers demonstrated no zonal differences in apoptosis. While the necrosis was only occasionally present in P14 LKO livers, a centrilobular (acinar zone 3) necrosis in livers of D24 AAV-Cre mice was observed. Please see the revised histological descriptions **in Page 7, Results section**. We performed additional IHC staining for RIPK3, a critical mediator of necroptosis, and confirmed increased RIPK3 expression concentrated in zone 3 of the livers of D24 AAV-Cre mice. We have added these data to **Supplemental Fig. S3a** in the revised manuscript.

3 – The impact of pancaspase inhibitor (e.g. Z-VAD) and RIPK1 inhibitor (Nec-1s) or both should be assessed to determine which cell death mechanism accounts for the liver failure.

Response: In response to this suggestion, we gave daily ip injections of a combination of Z-VAD-FMK and Necrostatin 1 (NEC1) at doses previously used in mice^{1,2}. The injections started at D10 in AAV-Cre mice, but the combined therapy failed to rescue lethality in AAV-Cre mice. This suggests that simply blocking the downstream hepatocyte apoptosis/necroptosis does not ameliorate neddylation deficiency-induced ALF. Please see **Supplemental Fig. S8h**. We also added discussion in **Paragraph 1 of Discussion** section.

4 - The authors show many pathologic and biochemical effects of inhibiting Neddylation. There is obviously a profound disturbance of hepatic metabolism making it difficult to pin-point the key events, though oxidative stress appears to account for some.

Response: We completely agree with the reviewer. Neddylation participates in diverse cellular processes and pathophysiological events. Our findings present the first line of evidence that neddylation is essential to normal liver function by regulating hepatocyte oxidative stress, mitochondrial dysfunction, NIK signaling etc. We think that dysregulation of these critical cellular processes collectively lead to a rapid development of fatal liver failure. Given the power of neddylation in liver, simply blocking one single pathway (NIK, TNF or cell death) may not be sufficient to rescue the fatal phenotype seen in NAE1-deficient mice. Our findings further demonstrate that antioxidants are the most effective treatment for ALF in AAV-Cre mice, similar to its efficacy in human ALF. This information have been included in **Paragraph 1, Discussion** section,

5 – The mechanism of oxidative stress certainly may be linked to GSH depletion and mitochondrial impairment but cause and effect are not clearly defined. The mechanistic link between NIK and the biochemical pathologic events needs to be explored more deeply. As presented this is not directly elucidated.

Response: In response to this comment, we performed proteomic analysis in HEK293T cells following co-expression of NIK-HA with FLAG-NEDD8, and identified four potential neddylation sites of NIK.

These sites were mutated and shown to be important in mediating NIK ubiquitination and protein stability (**new Fig. 6f-l**, and **Supplemental Fig. S7c-e**). We further demonstrated that NIK signaling contributes to neddylation deficiency-induced ALF both *in vitro* and *in vivo* using the NIK specific inhibitor B022. B022 alleviates MLN induced fetal reprogramming and desensitizes hepatocytes to TNF triggered apoptosis. We also administrated B022 daily to D10 AAV-Cre mice. A 50% inhibition of NIK signaling was achieved but this did not improve overall mortality in AAV-Cre mice. However, B022 treatment produced a trend towards reducing the expressions of progenitor and inflammatory genes and the necroptosis and apoptosis-related proteins in AAV-Cre livers. This suggests that NIK is involved in neddylation deficiency-induced acute hepatic injury. Detailed results are now included in **new Fig. 7f-h** and **Fig. S8g-h**. Our data therefore establish the first mechanistic link by which neddylation deficiency activates NIK signaling contributing to the pathogenesis of ALF.

Reviewer #2 (Remarks to the Author):

In this manuscript, the authors aim to clarify the role of neddylation deficiency in the regulation of acute liver injury. They reported that hepatic-specific inhibition of neddylation by NAE1 deletion caused hepatocyte damage, mitochondrial dysfunction, and excessive fetal liver reprogramming. Besides, the authors identified NIK as a novel NEDD8 target. The authors also reported that antioxidant therapy partially rescued NAE1 deficiency-induced ALF. Although the topic of the research and some of discoveries the authors presented are interesting, many of data are not solid enough to support the conclusions. Some of major issues that need to be addressed are listed below

1. The pattern of global protein Neddylation in multiple figures is quite different (e.g. Figs 1A/2A/5A/6A/6B/6D). In Figure 2A, there is a band above 55kD, while in the figure 5A, there is a band detected in 40kD. When NAE1 is downregulated, the neddylation of CULs should be inhibited. Therefore, neddylation modification of cullins, as the classic substrates of Neddylation, should be detected upon NAE1 downregulation.

Response: We thank the reviewer for making these important points. Other than CULs, the identity of neddylation substrates remain elusive. It is conceivable that the abundance of individual neddylated proteins varies between different cell types and at different developmental stages. The different migration patterns of neddylated proteins in different samples identified by the reviewer support this notion. Specifically, **Fig. 1a** and **2a** are from neonatal and adult livers, respectively; **Fig. 5a** is from isolated hepatocytes. In original **Fig. 6** now **Fig. 5**, **Fig. 5a** is from isolated primary hepatocytes infected with adenovirus, while **Fig. 5b** and **5d** are from HepG2 cells. we have now included neddylated CULs In each figure to confirm its downregulation due to NAE1 deletion or MLN inhibition (as indicated by arrow). We have also included neddylated CUL1 or CUL2 in each related panel to further validate the efficacy of NAE1 deletion or MLN inhibition on neddylation of CULs. Please see each relevant figure.

2. It is well known that neddylation deficiency causes mitochondrial dysfunction, apoptosis and cell death. The authors also showed that hepatocyte-specific neddylation deficiency caused hepatocyte damage, mitochondrial dysfunction, and excessive fetal liver reprogramming. Are the mechanisms of these phenotypes the same as the reported mechanisms of Neddylation inhibition, such as activating intrinsic and extrinsic apoptosis?

Response: MLN4924 has been reported to induce intrinsic (mitochondrial) apoptosis and activates the extrinsic (death receptor-mediated) apoptosis in cancer cells³. In our study, neddylation deficiency causes hepatocyte damage through induction of both intrinsic apoptosis due to mitochondrial dysfunction and extrinsic TNF-mediated apoptosis secondary to ROS or NIK mediated inflammation (**Fig. 4**). While the mechanisms underlying the neddylation deficiency-induced hepatocyte reprogramming remains to be further identified, a direct effect of neddylation in regulating NIK stability and function was elucidated (**Fig. 6 and Fig. 7**). We also discussed the potential mechanisms of mitochondrial dysfunction and GSH depletion in neddylation-deficient hepatocytes (Please see **Discussion page 19 and 20**).

3. It is well known that the inactivation of Neddylation-CRL pathway inhibited the degradation of I κ B α , leading to the inactivation of NF- κ B pathway. The authors showed the regulation of non-canonical NF- κ B pathway through hepatocyte-specific neddylation deficiency. How about the canonical NF- κ B pathway?

Response: We appreciate this insightful question. We have now included data on the canonical NF- κ B pathway (please see the revised **Fig. 7** and **Supplemental Fig. S8** as well as descriptions in **Page 15** of **Results** section). We found that NAE1 deletion produced a consistent activation of I κ B phosphorylation. However, the net change of canonical NF- κ B signaling was biphasic *in vivo*, with an early activation followed by a suppression (as evidenced by I κ B α degradation and p65 accumulation) during the ALF progression in AAV-Cre mice. *In vitro*, the degradation of I κ B α was not reliably inhibited in NAE1-knockdown primary hepatocytes or NAE1-deficient HepG2 cells. Therefore, neddylation deficiency leads to a consistent activation of non-canonical, but not of canonical NF- κ B signaling in hepatocytes, further emphasizing a direct role of neddylation-mediated NIK signaling in ALF.

4. In figure 6A, the authors claimed that they achieved \approx 50% NAE1 knockdown and suppression of neddylation in pAd-Cre. However, the N8-Conjugates had no obvious decrease upon NAE1 downregulation.

Response: We thank the reviewer for this question. Among the N8-Conjugates, we did not observe a reduction in the Cullin sizes (N8-CULs, which was confirmed by Western blot against CUL2) in NAE1 knockdown primary hepatocytes. This could be due to a 50% NAE1 knockdown being insufficient to

block neddylation of CULs. By contrast, we did observe that the neddylation conjugation of non-Cullin substrates was downregulated by NAE1 knockdown. Please see **new Fig. 5a**.

5. As mentioned by the authors, in figure 6B, in the NAE1-KO HepG2 cells, the N8-Conjugates strongly increased. The increase of N8-conjugates is quite confusing. The authors should detect the classic substrates of Neddylation (e.g. CUL1, CUL2) to further confirm the neddylation inhibition after knockdown of NAE1.

Response: We have performed Western blots against the classic neddylation substrates, such as CUL1, 2, and 3. We found that NAE1 knockout by two different gRNAs exerted differential effects on the neddylation of CUL1-3. While the neddylation of CUL1 was inhibited only in NAE1-KO2 HepG2 cells, reduced neddylation of CUL2 and CUL3 was obvious in NAE1-KO1 HepG2 cells (**Fig. R1**). Due to the limited space, we only included data for CUL2 (**new Fig. 5b**). Whether the increased signals at the Cullin sizes (N8-CULs) in NAE1-KO cell lines are due to changes in other CULs or are caused by nonspecific effects related to the CRISPR/Cas9 strategy remains unknown. We have now added this statement into the **legend** of **Fig. 5**.

6. In figure 6B, NAE1 knockdown is more efficient in KO2 compared with KO1. However, in figure S7C, the expression of NIK and p-p100 in KO1 is stronger than that in KO2. Meanwhile, the accumulation of p52 is not obvious enough.

Response: We appreciate the reviewer's insights. Currently we are not sure why activation of the NIK signaling pathway in two lines of NAE1 KO HepG2 cells are not dose dependent, which was the case for other signaling pathways we examined, such as YAP and hepatocyte reprogramming (**Fig. S6**).

7. In figure 7, the authors showed that NAE1 knockdown induced NIK accumulation, and MLN4924 treatment delayed the degradation of NIK (Fig. 7A-B). Co-treatment MLN and BZM synergistically elevated endogenous NIK protein accumulation (Fig. 7C). Therefore, the authors claimed that neddylation deficiency stabilizes NIK protein in a proteasome-dependent manner. All these results strongly indicate NIK may be a substrate of CRL E3 ligase, but not a direct substrate of neddylation. The authors should determine whether NIK is a substrate of CRL E3 ligase.

Response: We thank the reviewer for the valid points. NIK has been shown to be ubiquitinated by several non-CRL E3 ubiquitin ligases, such as cIAPs⁴, ZFP91 (Zinc finger protein 91)⁵, Peli1⁶ and CHIP

(carboxyl terminus of HSC70-interacting protein)⁷. To test whether CRL E3 ligases are involved, we obtained gRNAs against CUL1, 2, 3, 4a and 4b, respectively in an attempt to delete each CUL in HepG2 cells using a CRISPR/Cas9 strategy. We successfully obtained CUL2-KO and CUL3-KO HepG2 cells but there was no upregulation of NIK expression (these data have been included in **new Fig. S7a**).

To pinpoint whether NIK is a direct substrate of neddylation, we performed proteomic analysis and identified the potential neddylation sites of NIK. The involvement of these sites in NIK neddylation and protein stability was also examined (Please see response to Reviewer 1 comment 5). These data clearly strengthen our conclusion that NIK is a direct substrate of neddylation. Please see **the new Fig. 6** for more details.

8. In figure 7, the authors tried to prove that NIK is a neddylation substrate, however, the data provided are not sufficient to support this conclusion. Much more work is needed to strengthen the conclusion. Besides, in figure 7D, the Flag-NEDD8 band in input is very confusing. In figure 7E, the endogenous co-IP assay result is not satisfactory. I would say the interaction is so weak that it cannot be concluded that UBC12 interacted with NIK.

Response: We have added additional data to further prove that NIK is directly neddylated (please see response to Comment 7). In Fig. 7D (**now Fig. 6c**), we only presented the Western blot at the size of free NEDD8 form. While the FLAG-N8^{WT} could be largely conjugated to the substrates, the FLAG-N8^{AGG} was unable to conjugate, leading to the massive accumulation of FLAG-N8^{AGG}.

We have now scaled up the Co-IP assays in BZM-treated HepG2 lysates and identified a clear signal at the NIK size that co-immunoprecipitates with UBC12. These data provided direct evidence that endogenous NIK is a substrate of neddylation. Please see **Fig. 6d**.

9. The authors claimed that MLN dose-dependently induced post-transcriptional NIK protein expression and signaling in HepG2 cells (Figure S7, D-E) (the last lane of page 11 and first line of page 12). However, in figure S7E, the mRNA level of Mapk3k14 also seem to be upregulated, so this conclusion is not very clear. The authors did not provide the corresponding significance analysis in the bar plot or legend.

Response: Thank you for pointing that out. We have now added the significance analysis in the bar plot to show that the changes did not reach significance based on One-way ANOVA followed by Turkey's multiple comparisons test. Please see **now Fig. S8e**.

10. In Figure S6, the figure legend is not corresponding to the figure.

Response: We sincerely apologize for the mistake and have corrected the figure legend in **Fig. S6**.

11. In figure 3C, the C/EBP α is labeled P30. In figure 6A/6B/6D, the C/EBP α is labeled p42. What is the difference?

Response: Thank you for mentioning that. There are two isoforms of C/EBP α (p42 and p30) which play slightly different roles in liver development and function. We found that the antibody against C/EBP α (sc-61, scbt) dominantly detected the p30 form of murine source, whereas p42 was mainly detected in human samples such as HepG2 cells. Please see the source Western blot data with two different isoforms of C/EBP α labeled separately.

Reviewer #3 (Remarks to the Author):

NCOMMS-21-06607 Review

In this manuscript, the authors investigate a role for NAE1 and neddylation in human ALF disease. Through the use of conditional deletion of NAE1 in the embryo (Alb-CRE) and the adult knockouts, they show that reduction of neddylation leads to phenotypes similar to ALF, such as hepatocyte injury and death, liver inflammation and fibrosis. Gene Expression from both the embryonic knockout (LKO) and adult knockout show very similar responses in gene expression, as well as pathways that point increased oxidative stress and mitochondrial dysfunction which are thought to contribute to the liver phenotype.

They also show the decrease in mature hepatocyte gene expression and increase in genes and proliferation (Ki67+) reflecting BEC/progenitor following injury. This suggests a role for cellular reprogramming and/or regeneration in the pathophysiology of this phenotype. They also demonstrated a cell autonomous effect in CRISPR NAE1 knockouts in HepG2 cells and with neddylation inhibitor MLN in vitro with similar effects on gene expression.

Mechanistically, they show a connection between Neddylation and NF- κ B inducing kinase (NIK) stability. NIK is highly regulated by ubiquitination and degradation and is stabilized by a subset of TNF receptor family members upstream of alternative NF- κ B signaling. They show a similar stabilization and activation of NIK in the absence or inhibition of neddylation enzymes leading presumably to p100 phosphorylation and increased p52 processing. They also demonstrate a benefit with oxidative stress inhibition (NAC) on the mouse liver phenotype.

Comments:

(1) Only figure 6 refers to 3 independent experiments with representative images and plots of data. Would the authors please clarify how many times each set of observations was performed including embryonic and adult knockout cohorts?

Response: We have now included a statement on how many times of each *in vivo* and *in vitro* experiment was performed in the **Statistics Paragraph** of the **Methodology** section. We have also included specific animal numbers for each cohort experiment in each **Figure legend**.

(2) The authors suggest that there is significant perturbation of neddylation enzyme expression in two ALF patient datasets, with NAE1 being significantly downregulated. If they have access to ALF patient samples, measuring NAE1 expression for themselves would be preferable and would make a nice figure 1 A to set up the manuscript for why they have targeted NAE1. Authors may not have access to these human samples, but measuring NAE1 expression in multiple patient samples may raise the quality of the data and further support their reference to the phenotype in this disease as Figure 1A rather than in the supplement. Seeing the spread of the data in the patient population compared with control would be important.

Response: We agree with the reviewer that it would be preferable to measure NAE1 expression and neddylation levels in ALF patient samples. Unfortunately, we found no such commercial or academic source for these samples. Following the reviewer's suggestion, we have plotted the overall RPKMs of *NAE1* detected in the ALF patient populations from the two GSE datasets, which clearly demonstrated a reduction of *NAE1* expression in ALF groups. These data have been now included as **Supplemental Fig. S1b**.

(3) In figure 1 A, the authors show a nice reduction in N8-Conj in the KO P7 and p10, but this affect goes away by P14. Would the authors please comment on why this might be. Are their compensatory mechanisms that occur over time? Do other neddylation enzymes also use N8 as a substrate?

Response: We appreciate the reviewer's insight. We think that the lack of clear differences in N8-Conj in P14 LKO liver could be due to significant infiltration by non-hepatocytes. We have included this statement into the **Result** section, **Paragraph 2**.

(4) In figure 1, the liver injury, inflammation and fibrosis are clear. Curious whether they looked at hydroxyproline levels and/or collagen gene expression in the embryonic knockout? Collagen genes were measured in the adult knockout.

Response: We measured the expression of *Col1a1* and *Col1a3* genes by qRT-PCR and the results show a tendency toward higher expression in P14 LKO livers (Please see **Fig. 1i**). We also measured liver hydroxyproline levels. These were increased in livers of male P14 LKO mice (included as **Fig. 1j**), in line with the increased expression of fibrotic genes and elevated Sirius red staining. The hydroxyproline levels in adult D24 AAV-Cre livers were also higher compared to AAV-GFP mice. These new data are included in **Fig. 2g**.

(5) In reference to figure 1F, they refer to plasma ALT as a measure of liver function. While that is correct for plasma bilirubin, plasma ALT is a measure of hepatocyte damage and death not function.

Response: Thank you very much for pointing that out. We have reworded the text accordingly.

(6) The quality of the plot in figure 1H is poor. It is hard to see the dynamic range of the effect when the symbols are so low against the axis. This plot should be redone so that we can effectively see the fold change in expression of these genes.

Response: We have replotted **Fig. 1h** to make it more readable.

(7) Would the authors speculate on why they observe steatosis in the adult but not in the embryonic knockout?

Response: We have added in the **Paragraph 3, Discussion** at **page 20** that it could result from a more severe mitochondrial dysfunction observed in adult NAE1-deleted livers.

(8) In figure 3D, the reduction of mature hepatocyte markers and increase in BEC/progenitor markers is interesting. Is this due to mature hepatocyte death, reprogramming of some cells to progenitors or expansion of existing progenitors in the liver? It is hard to tell whether this effect is due to cellularity differences in the bulk tissue sample. Since this is RNA from bulk tissue, have the authors thought about doing single cell RNAseq to better tease out the cell populations that are present in these mice? The Ki67+ staining does suggest increased progenitor activation following injury, but given the current analysis it is hard to tell what they are and where they are coming from especially since the increase in progenitor activation (if it is occurring) does not keep up with the hepatocyte damage in this model. Is this an example of “frustrated” progenitors that are not capable of further differentiation following activation? Single cell RNAseq may better define known progenitor populations in the liver and give a more complete data set.

Response: We appreciate the reviewer’s brilliant proposal to perform single-cell RNA-sequencing to sort out the mechanisms underlying cell reprogramming in NAE1-deleted livers. ALF is characterized by a strong hepatic stem/progenitor cells gene signature, along with ductular reaction. An excessive reactivation of a fetal liver program is induced in hepatocytes when acutely injured livers regenerate⁸. The fetal reprogramming that occurs in AAV-Cre livers may include both hepatocyte reprogramming and progenitor activation in response to acute hepatocyte injury. While we agree it would be interesting to tease out the cell populations that are present in NAE1-deleted liver by scRNA-sequencing, we feel that this question takes us in a different direction than the focus of the current manuscript. Meanwhile, we encountered challenges in isolating sufficient viable hepatocytes in NAE1-deficient livers due to the rapid progression of disease. Therefore, we are currently constructing a lineage-tracing mouse line to specifically follow the fate of neddylation-deficient hepatocytes *in vivo*. This will help us more directly

address the question whether NAE1-deficient hepatocytes reprograms to hybrid hepatocytes or progenitors. Identifying the contributions of different cell populations in neddylation deficiency-induced fetal reprogramming will be the focus of our future study.

In addition, we performed co-staining of Ki67 with ALB in D24 AAV-Cre livers. The increase in % of Ki67⁺/Alb⁺ cells in AAV-Cre livers only accounts for a small fraction of the total Ki67⁺ cells, suggesting that both hepatocytes and non-hepatocytes are undergoing proliferation. These data are included in **new Fig. 3f**.

(9) Figure 4 shows a nice correlation between the embryonic and adult knockouts. Have the authors correlated these genes with the ALF data sets to see how relevant their model is to human AFL disease beyond just NAE1 expression and other neddylation enzymes? This would strengthen the relevance of the paper to human disease.

Response: Thank you very much for this excellent suggestion. We have compared the correlations of DEGs in AAV-Cre to AAV-GFP livers with the two ALF GSE datasets, respectively. As expected, their expression profiles demonstrated positive correlations. Meanwhile, most of DEGs related to oxidative stress and metabolic genes in AAV-Cre livers were dysregulated in both ALF GSE datasets (Please see **new Fig. 4c** and **page 9**). Together, our data demonstrate a clear pathological relevance of our model to certain types of ALF in humans.

(10) Figure 5 F Y-axis should start at 0 as not to overemphasize the effect between groups.

Response: We have replotted Y-axis of **Fig. 5F (now Fig. 4m)** accordingly.

(11) Figure 6 A, the authors should consider plotting C/ebpa and Hnf4a differently as to be able to see the fold change between groups. Although referred to as significant it is too small a difference as plotted to see effectively.

Response: We have replotted **Fig. 6A (now Fig. 5a)** accordingly. Thank you very much!

(12) Regarding Figure 6 A and B, would the authors please comment relatively limited effect in reduction of mature hepatocyte markers C/ebpa and Hnf4a. The effect between the two appears different between primary hepatocytes and HepG2 cells. This is less dramatic than the gene expression in bulk tissue. The authors should comment on why that might be and if this truly represents “reprogramming” to a progenitor like state or just transient gene expression differences. Again, single cell might give a more wholistic representation of what is going on here.

Response: In response to your comment, we have now included the following sentences in **Paragraph 5, Discussion** section at **page 21**.

“Whether the neddylation-deficient hepatocytes are fully reprogrammed to the progenitors or simply present as hybrid hepatocytes for liver regeneration *in vivo* remains unknown. Neddylation is hyperactivated in liver cancer⁹. Therefore, it is not surprising that neddylation deficiency causes more potent downregulation of hepatocyte marker proteins in protumoral as compared to primary hepatocytes (**Fig. 5a-b**). In contrast to a mild *in vitro* effect, neddylation deficiency leads to more severe reduction of mature hepatocyte marker genes in AAV-Cre livers *in vivo* (**Fig. 3**). Such a discrepancy could be attributed to the exacerbated hepatocyte loss triggered by initial hepatocyte injury-induced inflammation.”

(13) The authors clearly show the effect of neddylation on NIK stability. What is lacking in this dataset are the downstream effects of increased NIK stability/activity. The effect is mostly done with overexpressed NIK protein. Overexpression of NIK can have effects that may be different from endogenous levels of regulated NIK. Endogenous NIK can be hard to detect and pull down, but have they tried to look at the same effect on endogenous NIK? Have the authors thought about using proteomic approaches to observe neddylated and ubiquitinated NIK peptides?

Response: We thank the reviewer for the comments. We want to remind reviewer that the **Fig. 6a** and **6d**, **Fig. 7** and **Supplemental Fig. S8a-c** were all examining neddylation deficiency induced endogenous NIK protein expression and its downstream signaling pathway.

Human and murine NIK contains a total of 48 conserved lysine residues, 20 of them are allocated within N- 200 amino acids. Global high profiling Mass-spectroscopy analyses have identified 21 potential ubiquitination sites^{10, 11, 12}. In our study, we focused on identifying the potential neddylated peptides of NIK. We have performed proteomic analysis in HEK293T cells following co-expression of NIK-HA with FLAG-NEDD8. Please see response to Reviewer 1 comment 5. Detailed results are now included in the **new Fig. 6** and **Supplemental Fig. S7c-7e**.

(14) Figure 7A, the results appear inconsistent. Would the authors please comment on how many times these results were observed and how many individual mice were tested by western? The day 24 p52 blot is not very good. Is this whole tissue/cell lysate or nuclear extract? Nuclear extracts generally reflect a clearer increase in p52. Also curious what the total p100 bands look like above the p52 band. Do they observe a reduction in the p100 band given the rather modest effect on p52 in this case?

Response: In response to your comments, we have now examined NIK and its downstream signaling in whole tissue lysates (**new Fig. 7a**) and nuclear extracts (**new Supplemental Fig. S8a**) of another cohort of AAV-GFP and AAV-Cre animals at D21 and D24 post injection ($n = 3$ per group). The results show a clear elevation of NIK and its downstream signaling in AAV-Cre livers. These experiments were performed with a total of 6 mice (whole tissue lysates) and 3 mice (nuclear extracts).

We have now included both p100 and p52 in all figures involving NIK signaling. Interestingly, p100, a known ubiquitination substrate¹³, is also increased in neddylation-inhibited hepatocytes and livers. This suggests that neddylation deficiency also perturbs the protein expression of p100. However, p100 overexpression itself does not directly lead to its conversion to p52, which is more reliant on the activity of NIK and the downstream phosphorylation events¹⁴. Therefore, our findings unequivocally point to an elevated NIK-mediated, non-canonical NF- κ B signaling when neddylation is deficient.

(15) In figure 7, MLN inhibition leading to reduced neddylated NIK is not very strong. I do acknowledge that the IP western detection is not very quantitative, however the reduction in figure 7D & E is not very strong. Did the authors titrate the amount of MLN used and test earlier timing of the MLN treatment post HepG2 transfection to get a more optimal result? Does the residual neddylation represent the amount of NIK neddylated prior to MLN treatment? Would the authors speculate on the rate of turnover of neddylated NIK.

Response: We have now titrated the MLN doses and tested the time-dependent effects of MLN treatment on NIK neddylation (**new Fig. S7b**). We found that 0.5 μ M MLN and a 12 h treatment is most effective in inhibiting NIK neddylation. This suggests that the original dose and time points we performed in **Fig. 6c** are not optimal. However, we continuously observed an incomplete inhibition of NIK neddylation by MLN. It remains unknown whether the residual neddylation represents the amount of NIK neddylated prior to MLN treatment or the non-specific conjugation of overexpressed NEDD8 by the ubiquitin E1 enzyme¹⁵. Nevertheless, our data clearly show that NIK is a direct substrate of neddylation as MLN attenuates the neddylation of NIK. We also compared the turnover rates between wild-type NIK and the neddylation defective NIK K4R mutant, the latter is clearly much more stable after cycloheximide treatment. These data are included in **new Fig. 6i** and collectively highlight neddylation as a critical modulator of NIK stability.

(16) The authors make nice case for the effect of reduced oxidative stress (NAC treatment) on the NAE1 knockout phenotypes. Reduction in liver injury, inflammation, and fibrosis, gene expression as well as the mature hepatocyte vs BEC/progenitor genes, with clear controls to show reduction in markers of oxidative stress (supplement). What is lacking is the same rigor of working out the role of NIK in these phenotypes. Given the importance the authors are putting on the NIK neddylation connection, the same depth of investigation into the effect of NIK knockout or inhibition on the NAE1 / MLN phenotype in vivo would be important. The authors do point to the what has been shown in the literature about NIK transgene on liver injury / disease phenotypes, however, what is lacking is the direct effect of NIK inhibition / knockout on neddylation deficient phenotypes, such as hepatocyte injury and death, mature vs progenitor gene expression balance, oxidative stress, etc.

Response: Thanks for this suggestion. We administered B022 into AAV-Cre mice starting from D10 post virus injection. Please see response to Reviewer 1 comment 5 for details.

(17) Authors do show a 50% reduction in HepG2 apoptosis (Casp3) with NIK inhibitor B022 induced by TNF α similarly to NAC, but do not go beyond that. If the B022 compound is the one that I am familiar with from Amgen, the compound lacks significant potency as well as has selectivity limitations. Have the authors tried treating with B022 to see if NIK is required for the effect of NAE1 knockout or MLN on HepG2 induction of progenitor genes and repression of mature hepatocyte genes?

Response: We have performed the experiments in MLN-treated HepG2 cells and primary hepatocytes. In addition to a 50% reduction in TNF-induced HepG2 apoptosis by B022, our data shows that B022 can also partially suppress MLN-induced *KRT19* gene upregulation without restoring the reduced expression of hepatocyte marker genes (*C/EBPA* and *HNF4A*) (**new Fig. 7c**). B022 had no effect on MLN-induced upregulation of oxidative stress genes (*GCLC*) or downregulation of metabolic genes (*PCK1*) (**new Fig. 7c**), which was in contrast to NAC treatment in MLN-treated HepG2 cells (**new Fig. 8a**). Notably, B022 was also able to suppress *Krt19* upregulation in MLN-treated primary hepatocytes (**new Supplemental Fig. S8f**). These data are in line with the partial amelioration of progenitor gene activation in AAV-Cre livers by B022 (**new Fig. 7g**).

Overall, this manuscript demonstrates a clear effect of neddylation deficiency on liver injury and disease. The following should be addressed for acceptance:

(1) The authors show a reduction in mature hepatocyte markers and an increase in progenitor markers and suggest that there might be cellular reprogramming going on in response to liver injury. Rather than relying on bulk mRNA changes, the authors should use single cell RNAseq to identify cell populations that are present at the time of disease which would be greatly informative here.

Response: Please see response to Comment 8.

(2) A correlation between knockout RNAseq data sets with ALF correlated genes would greatly strengthen their story and the proposed connection between NAE1/Neddylation and ALF liver disease.

Response: Please see response to Comment 9.

(3) More correlation or efficacy data with NIK knockout or inhibition in the context of NAE1 knockout induced liver disease. While the NIK biochemical results on neddylation are intriguing, the proposed hypothesis of the relationship to human disease requires more direct links between NIK activity and downstream alternative NF- κ B signaling with neddylation induced disease and possible relationship to human ALF would be important to see if accepted for Nature Communications.

Response: Please see response to Comment 16.

REFERENCES

1. Linkermann A, *et al.* Rip1 (receptor-interacting protein kinase 1) mediates necroptosis and contributes to renal ischemia/reperfusion injury. *Kidney Int* **81**, 751-761 (2012).
2. Luk CT, *et al.* FAK signalling controls insulin sensitivity through regulation of adipocyte survival. *Nat Commun* **8**, 14360 (2017).
3. Chen P, *et al.* Neddylation Inhibition Activates the Extrinsic Apoptosis Pathway through ATF4-CHOP-DR5 Axis in Human Esophageal Cancer Cells. *Clin Cancer Res* **22**, 4145-4157 (2016).
4. Vince JE, *et al.* IAP antagonists target cIAP1 to induce TNFalpha-dependent apoptosis. *Cell* **131**, 682-693 (2007).
5. Jin X, Jin HR, Jung HS, Lee SJ, Lee JH, Lee JJ. An atypical E3 ligase zinc finger protein 91 stabilizes and activates NF-kappaB-inducing kinase via Lys63-linked ubiquitination. *J Biol Chem* **285**, 30539-30547 (2010).
6. Liu J, *et al.* Peli1 negatively regulates noncanonical NF-kappaB signaling to restrain systemic lupus erythematosus. *Nat Commun* **9**, 1136 (2018).
7. Jiang B, Shen H, Chen Z, Yin L, Zan L, Rui L. Carboxyl terminus of HSC70-interacting protein (CHIP) down-regulates NF-kappaB-inducing kinase (NIK) and suppresses NIK-induced liver injury. *J Biol Chem* **290**, 11704-11714 (2015).
8. Hyun J, Oh SH, Premont RT, Guy CD, Berg CL, Diehl AM. Dysregulated activation of fetal liver programme in acute liver failure. *Gut* **68**, 1076-1087 (2019).
9. Barbier-Torres L, *et al.* Stabilization of LKB1 and Akt by neddylation regulates energy metabolism in liver cancer. *Oncotarget* **6**, 2509-2523 (2015).
10. Akimov V, *et al.* UbiSite approach for comprehensive mapping of lysine and N-terminal ubiquitination sites. *Nat Struct Mol Biol* **25**, 631-640 (2018).
11. Kim W, *et al.* Systematic and quantitative assessment of the ubiquitin-modified proteome. *Mol Cell* **44**, 325-340 (2011).
12. Udeshi ND, *et al.* Refined preparation and use of anti-diglycine remnant (K-epsilon-GG) antibody enables routine quantification of 10,000s of ubiquitination sites in single proteomics experiments. *Mol Cell Proteomics* **12**, 825-831 (2013).
13. Chen ZJ. Ubiquitin signalling in the NF-kappaB pathway. *Nat Cell Biol* **7**, 758-765 (2005).
14. Xiao G, Harhaj EW, Sun SC. NF-kappaB-inducing kinase regulates the processing of NF-kappaB2 p100. *Mol Cell* **7**, 401-409 (2001).

15. Hjerpe R, *et al.* Changes in the ratio of free NEDD8 to ubiquitin triggers NEDDylation by ubiquitin enzymes. *Biochem J* **441**, 927-936 (2012).

Reviewers' comments:

Reviewer #2 (Remarks to the Author):

In the revised manuscript, the authors have addressed some of the concerns I mentioned last time. However, I would say that the some data present are still not convincing enough to draw a conclusion.

1. The authors claimed that neddylation deficiency upregulated NIK protein level, which was clearly demonstrated by the authors. The authors made efforts to confirm NIK to be a novel substrate of Neddylation and present new data in figure 6. However, the results provided in the revised manuscript are still not convincing enough to draw a conclusion that NIK is a direct substrate of neddylation.

In the revised manuscript, the authors performed proteomic analysis to identify the potential Neddylation sites of NIK. According to the methods and data provided by the authors (Fig. S7C-D), I think the identified peptides presented in Fig. S7D should be GlyGly modified peptides to characterize the potential Nedd8 modified peptides. However, these four modified peptides cannot be excluded as ubiquitinated peptides modified by endogenous ubiquitin (After trypsin digestion, both Neddylation and Ubiquitination would remain two glycine residues in the modified lysine of targets).

So far, the non-cullin substrates of Neddylation are very limited. The identification of novel non-cullin substrates of Neddylation is of great significance. In the recently published articles of non-cullin Neddylation substrates I found (PMID: 33299139 for PTEN Neddylation; PMID: 24821572 for Smurf1 Neddylation), several kinds of evidences have been provided to support the neddylation of novel targets. Therefore, the authors should provide more powerful evidence to support their conclusion.

2. In the response to comment 7 of the rebuttal letter, the authors used sgRNA strategies to knock out the CULs and successfully obtained CUL2-KO and CUL3-KO HepG2 cells. In Fig. S7A, the KO efficacy of CUL3 is not satisfactory. To determine if any of CULs is involved in regulating the stability of NIK, the authors are suggested to introduce siRNAs to knock down CULs or Rbx1/Rbx2 in HepG2 cells. That's a quick test compared to sgRNA strategy.

3. Some results the authors presented are not convincing enough, such as Fig. 5b. The authors explained that the increase signals of neddylation cullins in NAE-KO cells are caused by nonspecific effects of CRIPSR/Cas9 system. I would say the explanation given by the author is unacceptable.

Reviewer #3 (Remarks to the Author):

I thank the authors for the thoughtful and thorough response to the reviewer's questions. The authors improved the western blots addressing NIK and NFkB signaling and improved concerns of plot display and comments in the results and discussion sections. They also improved the comparison of knockout

DEGs with ALF data sets. The authors performed proteomic analysis linking NIK with specific sites of Neddylation and functionally linking Neddylation with NIK stability. This work represents an important contribution to the understanding of NIK regulation which is needed in the field.

Aside from a few comments to address, the authors have satisfactorily addressed my concerns for this manuscript to be published.

1. Figure S1b: I don't think plotting both data sets together is valid in this case. The two data sets should be plotted independently with statistics. This will give observation greater weight.
2. Figure 4: It is very nice to see the positive correlation between Knockout and ALF data sets. Although authors show the equation of the correlation line, pearson r and p-value would give greater statistical strength to the observation. Curious if the authors looked specifically at the strongest differentially expressed genes in the knockout data sets is the correlation stronger? Not needed for publication but could be interesting if the current overall positive correlation does not hit statistical significance.
3. Figure 7a / S8a: Authors suggest that there is an effect on p100 with Neddylation inhibition. The authors should add a comment that the possible increase in p100 is simply due to feedback loop with increased NIK stability and not necessarily directly linked to Neddylation.

Reviewer #4 (Remarks to the Author):

This study by Xu C et al investigates the role of neddylation in development of functional dysfunction in the liver and though the authors have extensively revised the manuscript based on earlier reviews, a few additional issues exist.

Major points

- 1) The resurgence of N8-Conjugated proteins (N8-Conj) in LKO mice by P14 is rather concerning and brings into consideration compensatory mechanisms. It would be useful to have immunostained liver sections to determine source of N8-Cul in P14 LKO, rather than speculating that this comes from infiltrating cells. This is all the more relevant since different hepatic cell types seem to be responding differently to modulation of neddylation since neddylation inhibition, by using the pharmacological inhibitor, MLN4924 was shown to reduce liver injury, apoptosis, inflammation, and fibrosis by targeting different hepatic cell types (PMID: 28035772).
- 2) Along these lines of compensatory mechanisms, it would be useful to evaluate the effects of changes in neddylation on ubiquitylation in some more detail. Though the authors allude to it in passing in the discussion, this is especially relevant since the pathways are inter-related for regulation of protein degradation and it is possible that compensatory mechanisms could be at play, especially since it is known that ubiquitin pathways regulate liver disease as well (PMID: 34529948) and interaction between the pathways has been recognized (PMID: 22608973).

3) The authors suggest that deficiency in neddylation compromises glutathione stores to induce oxidative stress and cell death, since supplementation with NAC protects against these changes to a certain extent. However, no mechanistic data is provided on why compromised neddylation should cause a decrease in cellular GSH. Some data to confirm mechanisms of this aspect would be useful.

4) The authors show higher steatosis in adult NAE1 deleted livers. However, it was recently demonstrated that inhibition of neddylation ameliorated steatosis in NAFLD (PMID: 34153521), which is opposite of what the authors indicate. This needs to be discussed and possible explanations provided.

5) The authors need to clarify what they interpret from the RIP3K data. Active caspase 8 would cleave RIP3K and induce apoptosis, while RIP3K activation in the absence of caspase 8 functions as a molecular switch for necrosis (PMID: 23913919). However, the authors show increase in both caspase 8 and RIP3K phosphorylation by D24 in Cre mice. However, there seems to be some individual variation since the animal with lower caspase 8 has higher RIP3K and vice versa. These issues need to be clarified.

6) What is the basis for the authors statement in the discussion that antioxidants prevent acute liver failure? While NAC is an antidote in drug induced liver toxicity such as due to acetaminophen, it functions by replenishing glutathione in that context, rather than scavenging free radicals. Also, NAC prevents liver injury if given early and is unlikely to prevent liver failure if injury is ongoing. The distinction between liver injury and liver failure needs to be made clear throughout the manuscript.

7) Page 3, Line 48-The authors state "The mechanisms underlying acute liver injury are not fully understood, especially at the posttranslational level." It is assumed the authors mean acute liver "failure" since mechanisms of injury are well characterized in several systems.

8) Page 5, Line 92: The authors state "This suggests the neddylation pathway may be involved in the pathophysiology of HBV-associated ALF." This sentence is not accurate, since the mere alteration of gene expression in microarray data sets is only that- an alteration. It does not confer a cause-effect confirmation and hence it the authors should modify the sentence.

9) Were the Nae1^{fl/fl} controls littermates? This needs to be clarified, and if so, specifically stated.

Responses to the Reviewers' Comments

Reviewer #2 (Remarks to the Author):

In the revised manuscript, the authors have addressed some of the concerns I mentioned last time. However, I would say that the some data present are still not convincing enough to draw a conclusion.

1. The authors claimed that neddylation deficiency upregulated NIK protein level, which was clearly demonstrated by the authors. The authors made efforts to confirm NIK to be a novel substrate of Neddylation and present new data in figure 6. However, the results provided in the revised manuscript are still not convincing enough to draw a conclusion that NIK is a direct substrate of neddylation.

In the revised manuscript, the authors performed proteomic analysis to identify the potential Neddylation sites of NIK. According to the methods and data provided by the authors (Fig. S7C-D), I think the identified peptides presented in Fig. S7D should be GlyGly modified peptides to characterize the potential Nedd8 modified peptides. However, these four modified peptides cannot be excluded as ubiquitinated peptides modified by endogenous ubiquitin (After trypsin digestion, both Neddylation and Ubiquitination would remain two glycine residues in the modified lysine of targets).

So far, the non-cullin substrates of Neddylation are very limited. The identification of novel non-cullin substrates of neddylation is of great significance. In the recently published articles of non-cullin Neddylation substrates I found (PMID: 33299139 for PTEN Neddylation; PMID: 24821572 for Smurf1 Neddylation), several kinds of evidences have been provided to support the neddylation of novel targets. Therefore, the authors should provide more powerful evidence to support their conclusion.

Response: We agree with this reviewer that identifying novel NEDD8 substrates is of great interest to the field. To date, addressing this critical question remains highly challenging, in part due to the relatively low abundance of endogenous neddylated substrates, the nature of the transient, dynamic modification, and inefficient enrichment of neddylated proteins, and as the reviewer pointed out, the difficulty in distinguishing neddylated proteins from ubiquitinated proteins. Nevertheless, we have managed to perform a series of experiments with a shortage of financial support and personnel. We have now collected substantial, compelling new evidence, which, along with those presented previously, collectively demonstrate NIK as a *bona fide* NEDD8 target. Below is the summary of evidence supporting NIK as a direct neddylation substrate (newly generated data were denoted as “**new**”).

- 1) Overexpression of WT-NEDD8, but not its conjugation-deficient counterpart, increased NEDD8-positive NIK species (**Fig. 7a**).
- 2) Inhibition of neddylation by pharmacological (MLN) and genetic (silencing UBC12) approaches reduced neddylated NIK (**Fig. 7a** and **new Fig. 7b**).

- 3) Overexpression of WT-SENPA8, a deneddylase, but not its catalytically inactive mutant (C163A), reduced neddylated NIK, while SENPA8 deletion or knockdown increases NIK neddylation (**new Fig. 7c-7d**).
- 4) Although current commercially available antibodies were not able to pull down endogenous NIK to validate its neddylation status *in vivo*, endogenous NIK binds to endogenous NEDD8 E2 enzyme UBC12 (**Fig. 7e**).
- 5) Mutation of all 49 lysines conserved in human and mouse to arginines (K49R) in NIK abolished NIK neddylation. Similarly, mutation of C-terminal 20 conserved lysines to arginines (K20R), but not the N-terminal 29 lysines to arginines (K29R), abrogated NIK neddylation (**new Fig. 7f-7g**).
- 6) Mass spectrometry analysis of NIK immunoprecipitates identified 4 lysines (K385/387/607/650) carrying di-glycine motif among C-terminal 20 lysines, which were enriched in cells co-overexpressing NIK and NEDD8, but not detectable in cells only expressing NIK (**new Supplemental Fig. S10a**).
- 7) Mutation of these 4 lysines to arginines reduced neddylated NIK, suggesting these 4 lysines are the acceptor sites for NEDD8 (**Fig. 7h**).

As this reviewer pointed out, we cannot rule out the possibility that the identified k-ε-GG peptides are modified by Ub due to the similarity of Ub and NEDD8 modification sites. Therefore, we have toned down the conclusion by stating these four lysines as “putative” neddylation sites.

In addition, we provided new evidence supporting the neddylation of NIK promotes its ubiquitination and degradation. We found impairment of neddylation by knockdown of UBC12 or NEDD8 and MLN significantly attenuated NIK ubiquitination (**New Fig 7i** and **Supplemental Fig. S9f**).

Figure 1 to Reviewers. a-b The neddylation of NIK-HA was examined by IB after IP under denaturing conditions (a) in 293T cells transfected with the indicated constructs and scrambled or DsiRNA against human cIAP1, and (b) in HepG2 cells transfected with the indicated constructs.

Accordingly, we have included more discussion on NIK neddylation and ubiquitination. Please see **Discussion, Paragraph 6 on Page 22**.

At last, we also made a great effort trying to identify the potential NEDD8 E3 ligase that mediates NIK neddylation. cIAP1/2 (cellular inhibitor of apoptosis 1 and 2) are the ubiquitin: NIK E3 ligases¹, and be NEDD8-E3 ligases of caspases². We found depletion of cIAP1 was not able to attenuate NIK neddylation. Overexpression of RBX1, another ubiquitin- and NEDD8- E3 ligase^{3,4},

⁵, did not promote NIK neddylation (**Figure 1 to Reviewers**). Therefore, the E3 ligase that mediates NIK neddylation remains to be identified. These negative data were not included in the main manuscript.

2. In the response to comment 7 of the rebuttal letter, the authors used sgRNA strategies to knock out the CULs and successfully obtained CUL2-KO and CUL3-KO HepG2 cells. In Fig. S7A, the KO efficacy of CUL3 is not satisfactory. To determine if any of the CULs is involved in regulating the stability of NIK, the authors are suggested to introduce siRNAs to knock down CULs or Rbx1/Rbx2 in HepG2 cells. That's a quick test compared to sgRNA strategy.

Response: We appreciate the constructive comments. Given the robust accumulation of NIK in neddylation-deficient cells and the liver, we agree that it is important to test whether neddylation regulates NIK stability through targeting cullins, the best-known NEDD8 targets. We found siRNAs yielded suboptimal knockdown efficiency in HepG2 cells due to low transient transfection efficiency. Therefore, we prefer to use CRISPR/Cas9. We deleted CUL1, CUL2, CUL3, or CUL4a in either HepG2 or NIH-3T3 cells by CRISPR/Cas9 and observed no impact on NIK expression (**new Supplemental Fig. S9a-S9d**). We have provided a better blot to show a significant, though incomplete, reduction of CUL3 in CUL3^{KO} cells. Moreover, we confirmed that deletion of CUL3 had no impact on NIK expression in livers of mice with hepatocyte-specific deletion of CUL3 (**Figure 2 to Reviewers**). Furthermore, deletion of RBX2, which pairs with CUL5 to form CRL5, in HepG2 cells did not alter NIK expression either (**new Supplemental Fig. S9e**). Despite multiple attempts, we failed to generate viable RBX1 knockout cell lines and therefore could not rule out a possible role for RBX1 in the regulation of NIK expression.

Together, our data do not support a critical role for CRLs in the regulation of NIK ubiquitination and stability. We have now included all these findings in the **new Supplemental Fig. S9a-e**. Please see the detailed description in **Results** Section on **Pages 14-15**.

3. Some results the authors presented are not convincing enough, such as Fig. 5b. The authors explained that the increased signals of neddylation cullins in NAE-KO cells are caused by nonspecific effects of the CRISPR/Cas9 system. I would say the explanation given by the author is unacceptable.

Figure 2 to Reviewers. Western blot in liver lysates from postnatal 14 WT and Cul3^{LKO} (*Cul3^{fl/fl}*, *Alb-Cre⁺*) mice. MLN-treated HepG2 cells were used as a positive control of NIK.

Figure 3 to Reviewers. Western blot analyses in HeLa cells infected with pLentiCRISPR/Cas9 V2 lentiviruses bearing two specific human NAE1 sgRNAs to generate bulk cultures with NAE1 knockout (KO1 and KO2), respectively. Cells infected with viruses expressing no sgRNA were used as control (V). Two independent experiments in duplicates.

Response: We agree that our original interpretation did not make sense. In **Fig. 5b**, the deletion of NAE1 was validated by a significant reduction of NAE1 proteins and neddylated CUL2. Currently, we do not know why NAE1 deletion in HepG2 by CRISPR/Cas9 leads to an upregulation of unknown NEDD8-positive species at the size of N8-CULs. Interestingly, we also observed elevated NEDD8-positive signals at ~45 KD in NAE1-KO HeLa cells by the same CRISPR/Cas9 strategy (**Figure 3 to Reviewers**). To avoid confusion, we have removed the NEDD8 blot from the original **Fig. 5b** and the related legends.

Reviewer #3 (Remarks to the Author):

I thank the authors for the thoughtful and thorough response to the reviewer's questions. The authors improved the western blots addressing NIK and NFkB signaling and improved concerns of plot display and comments in the results and discussion sections. They also improved the comparison of knockout DEGs with ALF data sets. The authors performed proteomic analysis linking NIK with specific sites of Neddylation and functionally linking Neddylation with NIK stability. This work represents an important contribution to the understanding of NIK regulation which is needed in the field.

Aside from a few comments to address, the authors have satisfactorily addressed my concerns for this manuscript to be published.

Response: We thank the reviewer for these very enthusiastic and positive comments on the quality and significance of our study.

1. Figure S1b: I don't think plotting both data sets together is valid in this case. The two data sets should be plotted independently with statistics. This will give observation greater weight.

Response: We appreciate the reviewer's suggestion. We have replotted the data and demonstrated that NAE1 expression was significantly lower in ALF patients as compared to healthy subjects in each dataset. Please see the **new supplemental Fig. S1b**.

2. Figure 4: It is very nice to see the positive correlation between Knockout and ALF data sets. Although authors show the equation of the correlation line, pearson r and p-value would give greater statistical strength to the observation. Curious if the authors looked specifically at the strongest differentially expressed genes in the knockout data sets is the correlation stronger? Not needed for publication but could be interesting if the current overall positive correlation does not hit statistical significance.

Response: We have now included the pearson r and p-values in **Fig. 4c**, which indicates statistical significance when compared to both datasets.

3. Figure 7a / S8a: Authors suggest that there is an effect on p100 with Neddylation inhibition. The authors should add a comment that the possible increase in p100 is simply due to feedback loop with increased NIK stability and not necessarily directly linked to Neddylation.

Response: We appreciate the reviewer's suggestion. However, we could not find evidence to support that increased NIK stability impedes partial p100 proteolysis to p52 via a feedback loop. NIK activates IKK α , and then the activated IKK α phosphorylates p100, leading to its polyubiquitination by the SCF-Fbw7-E3 ligase (CRL1) complex, and proteasome-mediated degradation to p52⁶. Therefore, we modified our sentence to **"The increase in p100 could be due to impaired CRL1-mediated p100 ubiquitination and degradation secondary to neddylation deficiency."** We have included this comment in **Results** on **Page 13** while describing **the original Fig. 7a, now Fig. 6a**.

Reviewer #4 (Remarks to the Author):

This study by Xu C et al investigates the role of neddylation in development of functional dysfunction in the liver and though the authors have extensively revised the manuscript based on earlier reviews, a few additional issues exist.

Major points

1. The resurgence of N8-Conjugated proteins (N8-Conj) in LKO mice by P14 is rather concerning and brings into consideration compensatory mechanisms. It would be useful to have immunostained liver sections to determine source of N8-Cul in P14 LKO, rather than speculating that this comes from infiltrating cells. This is all the more relevant since different hepatic cell types seem to be responding differently to modulation of neddylation since neddylation inhibition, by using the pharmacological inhibitor, MLN4924 was shown to reduce liver injury, apoptosis, inflammation, and fibrosis by targeting different hepatic cell types (PMID: 28035772).

Response: We have performed extensive double immunofluorescence staining of NEDD8 with hepatocyte marker Alb, BEC/progenitor marker Pan-CK, hepatic stellate cell (HSC) marker desmin, and liver macrophage (Kupffer cell) marker F4/80, respectively, in P14 WT and LKO livers. We found a specific reduction of Alb and its colocalized neddylation staining in hepatocytes, confirming hepatocyte-specific deletion of neddylation. However, we identified more Pan-CK and NEDD8 double-positive cells as well as more activated hepatic HSCs expressing desmin and NEDD8 in P14 LKO livers, whereas no significant differences in the staining of F4/80, a macrophage marker, and its co-localized NEDD8 were observed between two genotypes. These data confirmed the specific deletion of neddylation in hepatocytes and identified more NEDD8-positive BEC/progenitors and HSCs, which contribute to the recovery of total neddylation levels in P14 LKO livers. The increased number of Pan-CK and NEDD8 double-positive cells also further emphasizes the presence of fetal reprogramming in P14 LKO livers. We

have now presented these data in **new Supplemental Fig. S4** and described them in the **Results** section on **Page 8**.

2. Along these lines of compensatory mechanisms, it would be useful to evaluate the effects of changes in neddylation on ubiquitylation in some more detail. Though the authors allude to it in passing in the discussion, this is especially relevant since the pathways are inter-related for regulation of protein degradation and it is possible that compensatory mechanisms could be at play, especially since it is known that ubiquitin pathways regulate liver disease as well (PMID: 34529948) and interaction between the pathways has been recognized (PMID: 22608973).

Response: Neddylation is well known to target E3 ubiquitin ligases such as CRLs to mediate protein ubiquitination and degradation. Indeed, we have shown dysregulated NRF2 and I κ B α expression, two well-known CRL-targeted proteins, in our neddylation-deficient cells or tissues (**Fig. S5e** and **Fig. S8**). Interestingly, the total ubiquitination levels were upregulated in AAV-Cre livers. Whether such changes are due to the inactivation of deubiquitinating enzymes or alterations of cellular populations remains unknown. Nevertheless, we further assessed the expression of two additional CRL-targeted proteins including p53 and β -Catenin in our ALF model, which further supports our findings that neddylation deficiency dysregulates ubiquitin-mediated proteolysis, contributing to liver pathology. Data are now included as **New Supplemental Fig. S10i** and described in **Results** on **Pages 16-17**.

Meanwhile, PMID: 22608973 reports NEDD8 overexpression results in the neddylation of ubiquitin substrates by the ubiquitin pathway. This phenomenon could happen in many disease states when the NEDD8 level is elevated. Whether NAE1 deficiency causes elevated free NEDD8 leading to atypical neddylation of ubiquitination substrates to alter protein degradation is not known. A comprehensive study of proteome changes in neddylation-deficient livers would require quantitative proteomics, which is out of the scope of this study.

3. The authors suggest that deficiency in neddylation compromises glutathione stores to induce oxidative stress and cell death, since supplementation with NAC protects against these changes to a certain extent. However, no mechanistic data is provided on why compromised neddylation should cause a decrease in cellular GSH. Some data to confirm mechanisms of this aspect would be useful.

Response: We apologize for the confusion. In NAE1-deleted livers, the glutathione biosynthetic genes were not downregulated but elevated (**Fig. 4b** and **Supplemental Fig. S5**). This suggests that the reduced GSH level in NAE1-deleted livers is largely attributed to the increased oxidative stress through redox and conjugation reactions. Indeed, we found mitochondrial dysfunction and inflammation are among the mechanisms that cause oxidative stress in NAE1-deleted hepatocytes (**Fig. 4** and **Supplemental Fig. S5**). To avoid confusion, we have now reported GSH depletion after the increased

oxidative stress in **Fig. 4** and **Supplemental Fig. S5**. We have also modified our discussion accordingly (**Discussion, Paragraph 2**).

4) The authors show higher steatosis in adult NAE1 deleted livers. However, it was recently demonstrated that inhibition of neddylation ameliorated steatosis in NAFLD (PMID: 34153521), which is opposite of what the authors indicate. This needs to be discussed and possible explanations provided.

Response: We have now added discussion in **Discussion, Paragraph 3, and Page 20**. “Conversely, MLN has been shown to stimulate hepatocyte mitochondrial function and fatty acid oxidation to ameliorate steatosis in mouse models of dietary non-alcoholic fatty liver disease (NAFLD). The protective role of MLN was only observed under disease states with higher neddylation levels. Our current evidence suggests that neddylation, although relatively low in healthy adult hepatocytes, is essential for the maintenance of liver homeostasis under physiological conditions. Therefore, its level needs to be carefully fine-tuned when applying neddylation inhibition-based therapies to avoid potential adverse effects on liver function.”

5. The authors need to clarify what they interpret from the RIP3K data. Active caspase 8 would cleave RIP3K and induce apoptosis, while RIP3K activation in the absence of caspase 8 functions as a molecular switch for necrosis (PMID: 23913919). However, the authors show increase in both caspase 8 and RIP3K phosphorylation by D24 in Cre mice. However, there seems to be some individual variation since the animal with lower caspase 8 has higher RIP3K and vice versa. These issues need to be clarified.

Response: We thank the reviewer for pointing that out. Our RIPK3 staining identified necroptotic cells mainly concentrated in zone 3 (**Fig. S3a**), whereas apoptotic cells were scattered and azonal (**Fig. 2e**). Neddylation deficiency also did not trigger necroptosis cell-autonomously (Please see **Discussion, paragraph 4**). Therefore, caspase 8-mediated apoptosis and RIPK3-mediated necroptosis are occurring simultaneously in D24 Cre livers but not in the same cells. Since D24 Cre livers are very heterogeneous due to massive changes in liver cytostructures, individual variations are expected.

6. What is the basis for the authors statement in the discussion that antioxidants prevent acute liver failure? While NAC is an antidote to drug-induced liver toxicity such as due to acetaminophen, it functions by replenishing glutathione in that context, rather than scavenging free radicals. Also, NAC prevents liver injury if given early and is unlikely to prevent liver failure if injury is ongoing. The distinction between liver injury and liver failure needs to be made clear throughout the manuscript.

Response: We appreciate the reviewer’s comments. We understand the major role of NAC, as an antidote to drug-induced liver toxicity, is to replenish glutathione. However, the action of NAC results from its antioxidative or free radical scavenging property through increasing intracellular GSH levels. It can also act as a direct scavenger of free radicals. NAC is more effective in treating disease-associated

oxidative stress and inflammation^{7, 8, 9}. Especially, NAC treatment did replenish GSH, and reduce ROS and inflammation in AAV-Cre livers. Nevertheless, we have changed antioxidant therapy to NAC therapy.

We highly agree with the reviewer that NAC prevents liver injury if given early and is unlikely to prevent liver failure if injury is ongoing. We found exactly what the reviewer mentioned. NAC therapy is effective only if we provide NAC water at the early onset of NAE deletion (i.e. 10 days after AAV-Cre injection). Even if we start therapy 2 days later, NAC does not rescue mortality significantly. We have now included this information in the **Discussion, Paragraph 2**. We have also made distinctions between liver injury and liver failure throughout the manuscript.

7. Page 3, Line 48-The authors state “The mechanisms underlying acute liver injury are not fully understood, especially at the posttranslational level.” It is assumed the authors mean acute liver “failure” since mechanisms of injury are well characterized in several systems.

Response: Thank you very much for pointing that out. We have changed it accordingly.

8. Page 5, Line 92: The authors state “This suggests the neddylation pathway may be involved in the pathophysiology of HBV-associated ALF.” This sentence is not accurate, since the mere alteration of gene expression in microarray data sets is only that- an alteration. It does not confer a cause-effect confirmation and hence it the authors should modify the sentence.

Response: We have changed this sentence to “This suggests that the neddylation pathway is perturbed in HBV-associated ALF”. Thank you.

9. Were the *Nae1*^{fl/fl} controls littermates? This needs to be clarified, and if so, specifically stated.

Response: This was originally included in the **Results** section, **Paragraph 2** on **Page 5**. “*Nae1*^{fl/fl} mice were used as wild-type (WT) controls (**Fig. 1a**)”.

References:

1. Vince JE, *et al.* IAP antagonists target cIAP1 to induce TNFalpha-dependent apoptosis. *Cell* **131**, 682-693 (2007).
2. Broemer M, *et al.* Systematic in vivo RNAi analysis identifies IAPs as NEDD8-E3 ligases. *Mol Cell* **40**, 810-822 (2010).
3. Kamura T, Conrad MN, Yan Q, Conaway RC, Conaway JW. The Rbx1 subunit of SCF and VHL E3 ubiquitin ligase activates Rub1 modification of cullins Cdc53 and Cul2. *Genes Dev* **13**, 2928-2933 (1999).

4. Morimoto M, Nishida T, Nagayama Y, Yasuda H. Nedd8-modification of Cul1 is promoted by Roc1 as a Nedd8-E3 ligase and regulates its stability. *Biochem Biophys Res Commun* **301**, 392-398 (2003).
5. Ohta T, Michel JJ, Schottelius AJ, Xiong Y. ROC1, a homolog of APC11, represents a family of cullin partners with an associated ubiquitin ligase activity. *Mol Cell* **3**, 535-541 (1999).
6. Fukushima H, *et al.* SCF(Fbw7) modulates the NFkB signaling pathway by targeting NFkB2 for ubiquitination and destruction. *Cell Rep* **1**, 434-443 (2012).
7. Atkuri KR, Mantovani JJ, Herzenberg LA, Herzenberg LA. N-Acetylcysteine--a safe antidote for cysteine/glutathione deficiency. *Curr Opin Pharmacol* **7**, 355-359 (2007).
8. Dlodla PV, *et al.* N-Acetyl Cysteine Targets Hepatic Lipid Accumulation to Curb Oxidative Stress and Inflammation in NAFLD: A Comprehensive Analysis of the Literature. *Antioxidants (Basel)* **9**, (2020).
9. Zhang XF, *et al.* Conditional beta-catenin loss in mice promotes chemical hepatocarcinogenesis: role of oxidative stress and platelet-derived growth factor receptor alpha/phosphoinositide 3-kinase signaling. *Hepatology* **52**, 954-965 (2010).

REVIEWER COMMENTS

Reviewer #2 (Remarks to the Author):

The authors have extensively addressed my concerns and revised the manuscript based on earlier reviews. There are still some minor points in the revised manuscript. The authors are suggested to do a double check.

Line 212, β -TRCP should be Keap1.

Line 361 and 363, Fig 7i seems to be Fig 7j according to the authors' descriptions.

Reviewer #4 (Remarks to the Author):

The authors have responded to several concerns, but few issues persist.

1) It is unclear how the authors expect reactive oxygen species (ROS) to be detected on frozen liver sections with DHE, since these molecules by definition are highly reactive and unlikely to be preserved through sample collection, freezing and then incubation with reagents. Reliable measurement of ROS requires immediate measurement of these active species and hence the data in 4D should be removed. Moreover, the very mild changes in total ROS (Fig 4e), OCR (Fig 4k) and minor decrease in cell viability (Fig 4m) are unlikely to be biologically relevant as well. Hence the conclusion that oxidative stress is the primary cause of cell death in cells lacking NAE1 may not be accurate.

2) To make the case that GSH depletion was due to oxidative stress, the authors should also measure oxidized form of GSH which would concurrently increase with oxidant stress to confirm that this is the case. They mention that GSH was measured with a GSH/GSSG detection assay kit, but it is surprising that they do not show GSSG levels, which would be more pertinent to the argument for oxidative stress. A mere decrease in total GSH is not directly indicative of oxidative stress.

3) While it is true that NAC treatment increased total GSH levels slightly, as would be expected, it is unlikely to be the only mechanism of protection since levels do not come back to control levels. Based on the authors assumption that NAE-1 deficiency induced mitochondrial dysfunction and oxidative stress to induce downstream effects, confirmatory experiments using alternate mitochondrial targeted anti-oxidants such as Mito-Tempo could be used to confirm the findings.

Response to reviewers' comments

We thank the reviewers for their comments. We have now performed additional experiments and modified the text to address reviewers' concerns. Changes are highlighted in Red in the text.

Reviewer #2 (Remarks to the Author):

The authors have extensively addressed my concerns and revised the manuscript based on earlier reviews. There are still some minor points in the revised manuscript. The authors are suggested to do a double check.

Response: Thank the reviewer for the suggestion. We have performed more careful proofreading of the manuscript.

1) Line 212, β -TRCP should be Keap1.

Response: Thank the reviewer for pointing that out. We have corrected β -TRCP to KEAP1 in line 212.

2) Line 361 and 363, Fig 7i seems to be Fig 7j according to the authors' descriptions.

Response: We have corrected **Fig. 7i** on lines 361 and 363 to **Fig. 7j**. Meanwhile, we have also changed **Fig. 7j** on line 364 to **Fig. 7k**. Thank you very much for noticing that.

Reviewer #4 (Remarks to the Author):

The authors have responded to several concerns, but few issues persist.

1) It is unclear how the authors expect reactive oxygen species (ROS) to be detected on frozen liver sections with DHE, since these molecules by definition are highly reactive and unlikely to be preserved through sample collection, freezing and then incubation with reagents. Reliable measurement of ROS requires immediate measurement of these active species and hence the data in 4D should be removed. Moreover, the very mild changes in total ROS (Fig 4e), OCR (Fig 4k), and minor decrease in cell viability (Fig 4m) are unlikely to be biologically relevant as well. Hence the conclusion that oxidative stress is the primary cause of cell death in cells lacking NAE1 may not be accurate.

Response: DHE staining in frozen liver sections has been widely used to detect ROS in various liver diseases (Nat Commun. 2017, PMID: 29233977; Hepatology. 2018, PMID: 29251796; Cell Mol Gastroenterol Hepatol. 2019, PMID: 30576769; J Hepatol. 2019, PMID: 31295533; Hepatology. 2021, PMID: 32965675; Proc Natl Acad Sci U S A. 2021, PMID: 33468664; Hepatology. 2022, PMID: 35844150,

etc.). Therefore, our data in **Fig. 4d** is valid and added another direct evidence of the presence of oxidative stress in NAE1-deleted liver.

We want to emphasize that the total liver ROS was measured at the early stage (D21 post-virus injection) of disease progression. The OCR and cell viability assays were analyzed in primary hepatocytes isolated from livers at the initiation of NAE1 deletion (D12), which we originally discussed the potential reasons for their minor changes in **Discussion Paragraph 3**: “The mitochondrial dysfunction in cultured NAE1-deleted primary hepatocytes was probably underestimated given the selective attachment of relatively viable hepatocytes”. We also toned down by not claiming mitochondrial dysfunction as a primary cause of cell death in cells lacking NAE1 in **Results Paragraph 5 Lines 219 and 234**.

2) To make the case that GSH depletion was due to oxidative stress, the authors should also measure the oxidized form of GSH which would concurrently increase with oxidant stress to confirm that this is the case. They mention that GSH was measured with a GSH/GSSG detection assay kit, but it is surprising that they do not show GSSG levels, which would be more pertinent to the argument for oxidative stress. A mere decrease in total GSH is not directly indicative of oxidative stress.

Response: We thank the reviewer for the suggestion. We originally struggled with the Abcam kit (ab138881) which could only deduce the levels of GSSG by measuring the levels of total GSH+GSSG and GSH. We now obtained a GLUTATHIONE GSH/GSSG ASSAY KIT (MAK440, Sigma-Aldrich) which directly measures GSSG levels in the liver samples. As expected, we identified reduced GSH levels accompanied by increased GSSG levels and GSSG/GSH ratios, confirming the presence of oxidative stress in NAE1-deleted livers. Data now have been included in **Fig. 4f**.

3) While it is true that NAC treatment increased total GSH levels slightly, as would be expected, it is unlikely to be the only mechanism of protection since levels do not come back to control levels. Based on the authors assumption that NAE-1 deficiency induced mitochondrial dysfunction and oxidative stress to induce downstream effects, confirmatory experiments using alternate mitochondrial targeted anti-oxidants such as Mito-Tempo could be used to confirm the findings.

Response: We apologize for confusing the reviewer. Based on our data, we discussed in **Discussion Paragraph 1**, “Mechanistically, our data support the multiple-hit hypothesis which suggests neddylation deficiency simultaneously disrupts divergent signaling pathways, including but not limited to mitochondrial dysfunction, oxidative stress, hepatocyte reprogramming, and NIK activation”. Therefore, it is not appropriate for us to claim “Our data suggest that neddylation deficiency initiates the early phase of liver injury by inflicting mitochondrial damage”. We have now removed this sentence in **Discussion Paragraph 3**.

“NAC displays both direct and indirect antioxidant properties and is more effective in treating diseases associated with oxidative stress and inflammation^{41, 42, 44}” (**Discussion Paragraph 2 Line 431-433**). Mito-Tempo has protective effects on liver-related diseases. However, it is shown to induce secondary apoptosis during the late phase of APAP hepatotoxicity (PMID: 30324313). Meanwhile, multiple pathways (upregulated CYPs and inflammation as well as SOD1 downregulation, etc.) contribute to the oxidative stress in the NAE1 deletion-induced hepatotoxicity (**Fig. 4** and **Supplemental Fig. S5**). Therefore, long-term treatment of Mito-Tempo alone in our hepatic NAE1-deficient mice (at least from D10-D24) could be problematic and cannot be more effective than NAC to alleviate liver failure. Indeed, we found minimal-to-none effects when singly blocking TNF signaling, apoptosis/necroptosis, or NIK activation in our hepatic NAE1-deleted mice (**Supplemental Fig. S8**). Combined therapy with SOD1 overexpression, Mito-Tempo treatment, and/or NIK inhibitor could be promising but awaits further study in the future.

REVIEWERS' COMMENTS

Reviewer #4 (Remarks to the Author):

While the authors have responded to most of the queries, it is surprising that they insist on including the problematic measurements of ROS using DHE in cryo-sections. The authors provide no explanation on how they expect reactive oxygen species to survive tissue processing and should examine the consensus statement on guidelines for measuring reactive oxygen species and oxidative damage in cells and in vivo (PMID: 35760871), which clearly state that “ROS should not be ‘measured’ in tissue homogenates or cryosections, unless the probe or sensor employed is able to irreversibly capture the reactive species when the cells/tissues/organs are under biologically relevant conditions” Hence, the use of DHR to measure ROS in cryosections is not very meaningful and the data needs to be removed.

Response to reviewers' comments

We thank Reviewer 4 for the comments. We have now removed the problematic DHE staining. We also followed the editorial requests and revised the manuscript accordingly. Changes are highlighted in Red in the text.

Reviewer #4 (Remarks to the Author):

While the authors have responded to most of the queries, it is surprising that they insist on including the problematic measurements of ROS using DHE in cryo-sections. The authors provide no explanation on how they expect reactive oxygen species to survive tissue processing and should examine the consensus statement on guidelines for measuring reactive oxygen species and oxidative damage in cells and in vivo (PMID: 35760871), which clearly state that "ROS should not be 'measured' in tissue homogenates or cryosections, unless the probe or sensor employed is able to irreversibly capture the reactive species when the cells/tissues/organs are under biologically relevant conditions" Hence, the use of DHR to measure ROS in cryosections is not very meaningful and the data needs to be removed.

Response: We have removed all DHE-stained liver images in Fig. 4, Supplemental Fig. S5, and Fig. S11. We have toned down the mechanistic conclusion related to oxidative stress in Discussion, Paragraph 2.